# Learning to Reason Efficiently with Discounted Reinforcement Learning

**Alex Ayoub**[*]
Amazon, University of Alberta

**Kavosh Asadi**
Amazon

**Dale Schuurmans**
University of Alberta

**Csaba Szepesvári**
University of Alberta

**Karim Bouyarmane**
Amazon

## Abstract

Large reasoning models (LRMs) often consume excessive tokens, inflating computational cost and latency. More broadly, in goal reaching sequential decision problems we often want to reach the goal quickly, and LRM reasoning can be viewed through this lens. We challenge the assumption that longer responses improve accuracy. By penalizing reasoning tokens using a discounted reinforcement learning setup (interpretable as a small token cost) and analyzing Blackwell optimality in restricted policy classes, we encourage concise yet accurate reasoning, analogous to preferring shorter successful trajectories in a stochastic shortest path problem. Experiments confirm our theoretical results that this approach shortens chains of thought while preserving accuracy.

## 1 Introduction

Many sequential decision problems ask for policies that reach a goal quickly while maintaining a high probability of success. Large reasoning models (LRMs) increasingly solve math and code problems by emitting intermediate reasoning tokens before a final answer (Jaech et al., 2024), where the goal is a correct terminal response. Reinforcement learning (RL) post training (Sutton & Barto, 2018) improves accuracy but can lengthen responses (Liu et al., 2025a), raising inference cost and latency. Our objective is to train LRMs that reach correct answers effectively and efficiently: more concise reasoning with no loss in accuracy.

Longer chains of thought (Wei et al., 2022) are not free: they inflate compute and memory (quadratic attention and a growing key value (KV) cache), slow inference and reduce serving throughput. This makes the problem closely related to finding a stochastic shortest path (SSP): each additional reasoning token is an extra step, and we would like the shortest successful trajectory to a correct terminal state. Moreover, the role of length in accuracy is contested (Shao et al., 2024; Liu et al., 2025b; Lu et al., 2025; Fatemi et al., 2025) with many claiming there is an inherent tradeoff between length and accuracy. In this work we show that, up to a regime determined by the model class and problem instance, there is no tradeoff between accuracy and path length. Namely, one can reduce response length up to a certain instance dependent threshold without seeing a drop in accuracy. After the response length dips below this threshold, then accuracy begins to dip.

We model verifier based reasoning as a finite horizon Markov decision process (MDP) (Puterman, 2014) with a binary terminal reward, which can be viewed as a finite-horizon stochastic shortest path instance where the goal is terminal correctness and the path cost is reasoning length. We then train with a discount factor $\gamma < 1$. This design is motivated by Blackwell optimality (Blackwell, 1962; Puterman, 2014; Grand-Clément & Petrik, 2023): near $\gamma = 1$, discounting should preserve accuracy (goal reachability) while preferring shorter successful trajectories. In practice, we only apply discounting to the environment (correctness) reward. The amount of discounting depends only on reasoning length, leaving intrinsic formatting/shaping rewards undiscounted. Practically, we discount only reasoning tokens, regularize with a KL penalty to a moving reference policy (Peters

---

[*]Work done during internship at Amazon

et al., 2010) and ensure token budgets across methods are comparable for fair comparisons. Our contributions can be summarized as follows:

- Within any fixed (possibly restricted) policy class $\Pi$, we show that Blackwell optimal policies (optimal for all $\gamma$ sufficiently close to 1) *simultaneously* maximize undiscounted success (probability of reaching the correct terminal goal) and, among accuracy maximizers, minimize expected trajectory length (a stochastic shortest path criterion). Thus, up to a regime determined by the class, there is no tradeoff between accuracy and path length. Our result calls into question the claim that there is a tradeoff between accuracy and response length and establishes that one can shorten response length up to an instance dependent quantity as hypothesized by Lee et al. (2025)
- For finite $\Pi$, a Blackwell factor $\gamma_{\mathrm{bw}} < 1$ exists such that $\gamma$ optimal policies are constant for all $\gamma \in (\gamma_{\mathrm{bw}}, 1)$ and equal the Blackwell optimal set. We bound how close to 1 the discount must be to maintain accuracy while shortening average response length. This clarifies how to choose $\gamma$ when the deployment class is restricted.
- Using group relative policy optimization (GRPO) (Shao et al., 2024) with the discounted objective, we substantially reduce mean response length on GSM8K, MATH and additional math benchmarks while matching the undiscounted pass@1 baseline, in line with the shortest path prediction at fixed success probability.

Efficient goal-reaching (and, in particular, efficient reasoning) has been pursued via: (i) *RL with length based penalties*, which adds per token or per step penalties during policy optimization (Arora & Zanette, 2025; Su & Cardie, 2025; Ling et al., 2025; Xiang et al., 2025); (ii) *curated data approaches*, which fine tune on variable length or compressed traces to internalize concise reasoning (Fatemi et al., 2025; Hammoud et al., 2025; Qiao et al., 2025; Lu et al., 2025; Zhao et al., 2025; Shrivastava et al., 2025; Dai et al., 2025); and (iii) *prompt control*, which prompts the model to reason more concisely Aggarwal & Welleck (2025); Dumitru et al. (2025); Wu et al. (2025). We propose and analyze plain old discounting as a principled, instance aware mechanism. In finite horizon MDPs with binary terminal reward, maximizing the discounted correctness reward and minimizing expected path length coincide as the discount factor approaches one. Moreover, a small per step negative reward in this setting is equivalent to discounting (Bertsekas, 2012). See Sui et al. (2025) for a broader overview of efficient reasoning methods.

## 2 SETTING AND NOTATION

We model reasoning as a finite horizon discounted Markov decision process (MDP) (Puterman, 2014) which is given by the tuple $M = (\mathcal{S}, \mathcal{A}, P, r, H, \gamma, \mu)$. Here $\mathcal{S}$ and $\mathcal{A}$ are finite state and action spaces, $P : \mathcal{S} \times \mathcal{A} \to \Delta(\mathcal{S})$ is the transition kernel, $r : \mathcal{S} \times \mathcal{A} \to \mathbb{R}$ is a bounded reward (verifier), $H \in \mathbb{N}$ is the horizon, $\gamma \in [0, 1)$ is the discount factor,[1] and $\mu \in \Delta(\mathcal{S})$ is the distribution over initial states (questions) where $\Delta(\mathcal{S})$ is the set of probability distributions over states.

A (possibly nonstationary) policy $\pi = (\pi_t)_{t=1}^{H}$ consists of maps $\pi_t(\cdot \mid s) \in \Delta(\mathcal{A})$ for each $t$. Fixing the start state, $s$, a policy (or language model) induces a distribution $\mathbb{P}_{\pi,s}$ over trajectories

$$S_1, A_1, R_1, \ldots, S_H, A_H, R_H, S_{H+1}, \quad A_t \sim \pi_t(\cdot \mid S_t), \ R_t = r(S_t, A_t), \ S_{t+1} \sim P(S_t, A_t).$$

The (discounted) state value function of $\pi$ is

$$v_\gamma^\pi(s) = \mathbb{E}_{\pi,s} \left[ \sum_{t=1}^{H} \gamma^{t-1} R_t \right],$$

where $\mathbb{E}_{\pi,s}$ is the expectation corresponding to $\mathbb{P}_{\pi,s}$. The $\mu$ weighted return is

$$J_\gamma(\pi) = \int v_\gamma^\pi(s) \, \mu(ds).$$

---

[1] We use $\gamma \in [0, 1)$ for analysis; when defining *accuracy* we also consider $\gamma = 1$ in finite horizon.

## 2.1 LANGUAGE MODELING

In language modeling, actions are vocabulary tokens and states are token sequences. The next state is the current sequence with the chosen token appended:

$$S_{t+1} = P(S_t, A_t) = S_t A_t$$

where we write $xy$ for the concatenation of $x$ and $y$. The special action eos ends the episode and moves to an absorbing terminal state. After taking eos, the process remains in an absorbing state with zero reward for the remainder of the horizon. If eos is not emitted by time $H$, we deterministically transition to a terminal state that triggers the verifier.

In the so called RL with verifiable rewards (RLVR) (Lambert et al., 2024), the verifier returns $1$ if and only if the sequence at emission of eos contains a correct final answer and $0$ otherwise:

$$r(S_t, \text{eos}) = \mathbb{I}\{S_t \text{ contains a correct answer}\}, \qquad r(S_t, a) = 0 \text{ for } a \neq \text{eos}.$$

Under this reward, the undiscounted finite horizon return equals the success probability. We therefore define the (Pass@1) accuracy of $\pi$ as

$$\text{Acc}(\pi) := J_1(\pi) = \int \mathbb{P}_{\pi,s}\big(\text{correct within } H\big)\, \mu(ds),$$

i.e., the fraction of prompts (under $\mu$) for which the first generated solution is verified correct.

## 3 BLACKWELL OPTIMALITY AND OUR MAIN THEORETICAL RESULTS

To formalize maximizing accuracy while minimizing mean response length, we use a stronger notion of optimality than is standard in reinforcement learning: the notion introduced by Blackwell (1962), henceforth Blackwell optimality (Puterman, 2014; Grand-Clément & Petrik, 2023). A policy is Blackwell optimal if it is optimal for all discount factors sufficiently close to one. This is relevant because the optimal policy in goal reaching MDPs (RLVR) at $\gamma = 1$ maximizes success (accuracy), while—as we show below—the optimal policy for $\gamma < 1$ is the one that reaches the goal via the shortest path. If a policy is optimal both for $\gamma < 1$ (near one) and for $\gamma = 1$, then it simultaneously maximizes accuracy and minimizes mean response length. *The missing proofs of all our results can be found in our appendix.*

**Why Blackwell optimality?** Discounting with $\gamma < 1$ breaks ties between equally successful policies by preferring earlier success, but if $\gamma$ is not sufficiently close to $1$ it may instead prefer a shorter yet less successful policy. The following example, with a restricted stochastic three-policy class, illustrates both effects. We consider restricted stochastic policy classes as this is a simplified model of softmax policy classes which are standard when analyzing policy gradient methods (Sutton & Barto, 2018; Mei et al., 2020).

**Proposition 3.1.** *Fix $p \in (0,1)$ and $0 < q_1 < q_2 < 1$, and consider the MDP in Figure 1 with horizon $H \geq 4$ and deterministic initial state $s_0$. Let the restricted policy class be $\Pi = \{\pi_1, \pi_2, \pi_3\}$, where at $s_0$: for $i \in \{1, 2\}$, $\pi_i$ selects $a_3$ with probability $q_i$ and $a_2$ with probability $1 - q_i$, and $\pi_3$ selects $a_1$ with probability $p$ and $a_4$ with probability $1 - p$. Let $\tau(\pi)$ denote the time step at which $a_{\text{end}}$ is taken under policy $\pi$. Then*

$$J_1(\pi_1) = J_1(\pi_2) = 1, \quad \mathbb{E}[\tau(\pi_i)] = 3 + q_i \ (i = 1, 2), \quad J_1(\pi_3) = p, \quad \tau(\pi_3) = 2.$$

*For all $\gamma \in [0, 1)$,*

$$J_\gamma(\pi_i) = (1 - q_i)\gamma^2 + q_i\gamma^3 \quad (i = 1, 2), \qquad J_\gamma(\pi_3) = p\gamma.$$

Thus there exists a threshold $\gamma' \in (p, 1)$ such that for every $\gamma > \gamma'$, $\pi_1$ is both an optimal policy in $\Pi$ and a shortest path policy. This example motivates Blackwell optimality: it selects the shortest policy among success maximizers (as $\gamma \uparrow 1$), while excluding policies that become optimal only by sacrificing success probability at smaller $\gamma$.

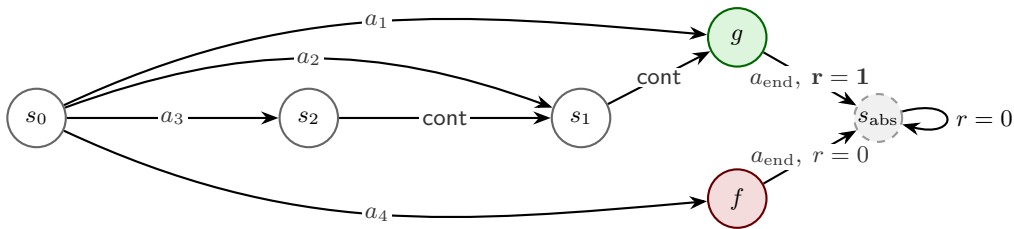

Figure 1: A finite-horizon MDP illustrating the conflict between success probability and discounting. **Green** ($g$) indicates the goal state ($r = 1$), while **Red** ($f$) indicates failure ($r = 0$).

**Roadmap of the theory.** Our analysis starts from a uniform Taylor expansion of $J_\gamma(\pi)$ around $\gamma = 1$, which makes explicit the stochastic shortest path (SSP) structure induced by discounting: maximize success probability and, among maximizers, minimize successful path length. We then relate this SSP objective to Blackwell optimality, and finally establish existence of Blackwell optimal policies for restricted finite policy classes.

We now introduce the formal definition of a Blackwell optimal policy. Recall that we assume finite horizon $H < \infty$, finite state and action sets, and bounded rewards.

**Definition 3.2.** Given $\gamma \in [0, 1)$, a policy $\pi \in \Pi$ is $\gamma$ discount optimal if $J_\gamma(\pi) \geq J_\gamma(\pi')$ for all $\pi' \in \Pi$. We call $\Pi_\gamma^\star \subset \Pi$ the set of $\gamma$ discount optimal policies.

**Definition 3.3** (Blackwell (1962)). A policy $\pi$ is Blackwell optimal if there exists a $\gamma \in [0, 1)$ such that $\pi \in \Pi_{\gamma'}^\star$ for all $\gamma' \in [\gamma, 1)$. We call $\Pi_{\text{bw}}^\star$ the set of Blackwell optimal policies.

Note that our definition of optimality is with respect to both an MDP instance $M$ and a policy class $\Pi$, whereas the usual notions of optimality (and the existence of an optimal policy) (Puterman, 2014; Bertsekas, 2019; Szepesvári, 2022) depend only on the MDP $M$.

### 3.1 MAIN THEORETICAL RESULTS

We now specialize to the setting of reaching a goal (RLVR) where reward is a deterministic binary terminal verifier.

**Assumption 3.4.** There exists a termination action $a_{\text{term}} \in \mathcal{A}$ (e.g., eos), an absorbing state $s_{\text{abs}} \in \mathcal{S}$, and a goal set $G \subseteq \mathcal{S}$ such that for all $s \in \mathcal{S}$:

1. $r(s, a) = 0$ for all $a \neq a_{\text{term}}$;

2. taking $a_{\text{term}}$ transitions to the absorbing state, i.e. $P(s_{\text{abs}} \mid s, a_{\text{term}}) = 1$;

3. the terminal reward is deterministic and binary, $r(s, a_{\text{term}}) = \mathbb{I}\{s \in G\} \in \{0, 1\}$. Moreover, the absorbing state yields no further reward and transitions to itself: for all $a \in \mathcal{A}$,

$$r(s_{\text{abs}}, a) = 0, \qquad P(s_{\text{abs}} \mid s_{\text{abs}}, a) = 1.$$

Let $\tau \leq H$ be the (first) absorption time. Define the success probability and (conditional) successful path length

$$p(\pi) = \mathbb{P}_{\pi,\mu}(\text{success within } H), \qquad L(\pi) = \mathbb{E}_{\pi,\mu}[\tau \mid \text{success}],$$

with the convention that the product $p(\pi)(L(\pi) - 1)$ is interpreted as 0 when $p(\pi) = 0$. Call $\pi$ a *shortest path policy* if it maximizes $p(\pi)$ and, among all maximizers of $p$, minimizes $L(\pi)$. If $p_\star := \max_\pi p(\pi) = 0$, the shortest path condition reduces to the first criterion.

**Lemma 3.5** (Uniform Taylor expansion). *Let $\varepsilon = 1 - \gamma$. Under Assumption 3.4, for every policy $\pi$,*

$$J_\gamma(\pi) = \mathbb{E}_{\pi,\mu}\big[\gamma^{\tau-1}\mathbf{1}\{\text{success}\}\big] = p(\pi)\Big(1 - \varepsilon(L(\pi) - 1)\Big) + R_\pi(\varepsilon),$$

*with remainder satisfying the uniform bound $|R_\pi(\varepsilon)| \leq C_H \varepsilon^2$, where $C_H := \frac{1}{2}(H - 1)(H - 2)$.*

Lemma 3.5 makes the SSP structure explicit: as $\gamma \uparrow 1$, the leading term is $p(\pi)$ (accuracy), and discounting breaks ties among success maximizers using the next term, which depends on $L(\pi)$ (successful path length). In other words, for $\gamma$ sufficiently close to 1, maximizing $J_\gamma$ implements the SSP objective "maximize success probability, then minimize (conditional) steps to success" (Bertsekas, 2012; Puterman, 2014). We now show that Blackwell optimality recovers the shortest-path SSP objective in goal reaching MDPs (RLVR).

**Theorem 3.6.** *In finite-horizon MDPs with a deterministic binary terminal verifier reward (Assumption 3.4), every Blackwell optimal policy is a shortest path policy:*

$$\Pi_{\mathrm{bw}}^\star \subseteq \operatorname*{argmin}_{\pi \in \Pi_{\max p}} L(\pi), \quad where \quad \Pi_{\max p} = \operatorname*{argmax}_{\pi \in \Pi} p(\pi).$$

Theorem 3.6 give our main theoretical guarantee. Namely that Blackwell optimal policies are accuracy maximizing and have the shortest mean response length within the class of accuracy maximizing ($\gamma = 1$) policies.

**Proof intuition (sketch).** Taking the Taylor expansion (Lemma 3.5) of $J_\gamma(\pi^\star) - J_\gamma(\pi)$ gives

$$p(\pi^\star) - p(\pi) - (1-\gamma)(p(\pi^\star)(L(\pi^\star) - 1) - p(\pi)(L(\pi) - 1)) + O\big((1-\gamma)^2\big) \ .$$

Since $\pi^\star$ is Blackwell optimal it must be optimal for all $\gamma$ arbitrarily close to one. Thus if some $\pi$ had $p(\pi) > p(\pi^\star)$, the leading term $p(\pi^\star) - p(\pi) < 0$ would make $J_\gamma(\pi^\star) - J_\gamma(\pi) < 0$ for $\gamma$ close enough to 1, contradicting Blackwell optimality (Definition 3.3). Therefore $\pi^\star \in \arg\max_{\pi \in \Pi} p(\pi)$. Moreover, among policies with $p(\pi) = p(\pi^\star)$, the first term cancels and optimality for $\gamma \to 1$ forces $L(\pi^\star) \leq L(\pi)$.

**Existence of Blackwell optimality for restricted classes.** The SSP characterization above is useful only if Blackwell optimal policies exist in the restricted class $\Pi$. We now adapt classical Blackwell arguments (Blackwell, 1962; Zwick & Paterson, 1996; Puterman, 2014; Grand-Clément & Petrik, 2023) to the case where the admissible class is restricted.

**Assumption 3.7** (Finite policy class). The admissible class $\Pi$ is finite: $|\Pi| < \infty$.

**Theorem 3.8.** *Given a finite horizon MDP $M$, under Assumption 3.7, there exists $\gamma' \in [0,1)$ and a nonempty set $\Pi_{\mathrm{bw}}^\star \subseteq \Pi$ such that for all $\gamma \in (\gamma', 1)$,*

$$\operatorname*{argmax}_{\pi \in \Pi} J_\gamma(\pi) = \Pi_{\mathrm{bw}}^\star.$$

Combining Theorem 3.8 with Theorem 3.6 yields that, in goal reaching MDPs (RLVR) with a finite restricted policy class, there exists a policy that is discounted optimal for all $\gamma$ sufficiently close to 1, and this policy is a shortest path policy in the sense above. We now introduce the Blackwell discount factor, first introduced by Grand-Clément & Petrik (2023).

**Definition 3.9.** The Blackwell discount factor is

$$\gamma_{\mathrm{bw}} := \inf \left\{ \gamma \in [0,1) : \ \Pi_{\gamma'}^\star = \Pi_{\mathrm{bw}}^\star \ \forall \gamma' \in (\gamma, 1) \right\},$$

where $\Pi_\gamma^\star = \arg\max_{\pi \in \Pi} J_\gamma(\pi)$.

At a high level, the Blackwell discount factor $\gamma_{\mathrm{bw}}$ guarantees that any policy that is discount optimal for $\gamma \in [\gamma_{\mathrm{bw}}, 1)$ is also Blackwell optimal. This reduces finding a Blackwell optimal policy to solving for a discount optimal policy.

**Instance hardness and a uniform critical discount.** The quantity $\gamma_{\mathrm{bw}}$ is instance dependent since it depends on the MDP instance $M$ and on the admissible class $\Pi$. This quantity captures how "hard" it is for discounting to reliably implement the SSP tie breaking without sacrificing success probability. Concretely, the closer $\gamma_{\mathrm{bw}}$ is to 1, the smaller the margin separating the best SSP behavior from near-ties in $\Pi$, and the more carefully one must choose $\gamma$ to avoid preferring shorter but less accurate behaviors.

Crucially, in our setting each reasoning instance induces a finite horizon MDP, so $\gamma_{\mathrm{bw}}(M, \Pi) < 1$ for every instance. Moreover, for any finite collection of instances $\{M_i\}_{i=1}^N$, one may define a single critical discount factor

$$\gamma_{\mathrm{crit}} := \max_{i \in [N]} \gamma_{\mathrm{bw}}(M_i, \Pi) \ < \ 1,$$

and then any $\gamma \in (\gamma_{\mathrm{crit}}, 1)$ is simultaneously "close enough to 1" for all those instances. In this sense, finiteness of the problem family yields a single discount choice that preserves instance-wise SSP optimality across the entire finite set. We now state a result that shows that for an arbitrary finite restricted policy class $\Pi$, the Blackwell discount factor exists.

**Lemma 3.10.** *Given a finite horizon MDP $M$, under Assumption 3.7, the Blackwell factor $\gamma_{\mathrm{bw}}$ exists and satisfies $\gamma_{\mathrm{bw}} < 1$.*

*Proof.* Theorem 3.8 ensures that $\Pi_\gamma^\star$ is constant for all $\gamma$ sufficiently close to 1, so the infimum in Definition 3.9 is well defined and strictly less than 1. □

The next lemma establishes that for finite horizon problems, a Blackwell optimal policy must also be optimal for the undiscounted objective.

**Lemma 3.11.** *A Blackwell optimal policy is also optimal in the undiscounted problem.*

*Proof.* Suppose $\pi$ is Blackwell optimal: $\pi \in \Pi_{\mathrm{bw}}^\star$. Then for any policy $\pi'$ we have $J_\gamma(\pi) - J_\gamma(\pi') \geq 0$ for all $\gamma \in [\gamma_{\mathrm{bw}}, 1)$. Therefore since $J_1(\pi)$ is well defined for finite horizon MDPs,

$$\lim_{\gamma \to 1} J_\gamma(\pi) - J_\gamma(\pi') \geq 0 \,.$$

We also know that $J_\gamma(\pi) - J_\gamma(\pi')$ is a polynomial and therefore continuous. Thus, it must be that $J_{\gamma=1}(\pi) - J_{\gamma=1}(\pi') \geq 0$, i.e. $\pi$ is also optimal in the undiscounted problem. □

### 3.2 Softmax training, greedy deployment

We now consider a common setting in language model post training and deep reinforcement learning where we use softmax policies for training and then evaluate (or deploy) the greedified policy (Haarnoja et al., 2018). This setting is important as our experimental setup will train softmax policies and evaluate their greedified variants. We fix a deterministic tie breaking rule on $\mathcal{A}$ and define the greedification map on the states

$$\mathrm{Greed}(\pi, s) \in \arg\max_{a \in \mathcal{A}} \pi(a \mid s) \qquad \forall s \in \mathcal{S} \,.$$

The deployment class is the image $\Sigma := \{\mathrm{Greed}(\pi, \cdot) : \pi \in \Pi_s\}$, a subset of the deterministic stationary policies on the finite horizon MDP $M$. Each $\sigma \in \Sigma$ corresponds one-to-one to a deterministic nonstationary policy. In our appendix, we also provide a bound on the Blackwell discount factor of the policy class $\Sigma$ for completeness, see Theorem A.12 for more details.

## 4 Training methodology

Guided by the theory in the previous section, we translate discounting into a practical training recipe for efficient reasoning with language models. Our design has four components:

1. **Discount only the environment (correctness) reward.** We apply a discount factor $\gamma \in (0, 1)$ to the environment reward but not to the learner's intrinsic formatting/shaping reward. This preserves the incentive to produce well structured outputs while encouraging shorter, more efficient chains of reasoning.

2. **KL regularization to a changing reference policy.** We use KL regularization against a reference model that is updated over training, following standard practice in policy gradient methods (Peters et al., 2010; Mei et al., 2020; Vieillard et al., 2020; Vaswani et al., 2022). This viewpoint aligns with relative entropy policy search (Peters et al., 2010) and has also been adopted in recent language model alignment work (Gorbatovski et al., 2025).

3. **Discount only reasoning tokens.** Discounting is applied exclusively to tokens used for reasoning; we do not discount tokens required for prompt adherence, formatting, or final answer presentation.

4. **Comparable token budgets across methods.** To ensure fairness, we make token budgets across methods comparable: since discounting shortens reasoning traces, we increase the number of rollouts for discounted methods so that the total tokens processed—and hence training accuracy—are comparable to the undiscounted baseline.

**Objective.** Because both the correctness and formatting signals are computed only at the end of the trajectory, we use a sequence level return. Let $m_t \in \{0, 1\}$ indicate whether token $t$ is part of the reasoning span and define the number of reasoning tokens $K(\tau) \triangleq \sum_t m_t$. Let $r^e(\tau)$ be the environment/correctness reward and $r^f(\tau)$ the formatting/shaping reward, both evaluated at the end of the rollout $\tau$. We discount only the environment reward as a function of reasoning length:

$$R(\tau) = \gamma^{K(\tau)} r^e(\tau) + r^f(\tau). \tag{1}$$

The learner then optimizes

$$\mathbb{E}_{S_1 \sim \mu, \, \tau \sim \pi(S_1)} [R(\tau)] - \beta \operatorname{KL}(\pi \mid \pi'), \tag{2}$$

where $\pi'$ is a reference policy that changes over training (defined below) and $\beta > 0$ sets the regularization strength. Equation (1) applies discounting only through $K(\tau)$, leaving formatting tokens undiscounted, in accordance with the Blackwell optimality perspective.

**Implementation details.** (i) *Reasoning mask.* The indicator $m_t$ isolates tokens that perform latent computation (chain of thought or tool use) from tokens required for formatting or final answer emission. (ii) *Reference updates.* The reference $\pi' = \pi_{\text{ref}}^{(u)}$ is updated periodically (e.g., at epoch or fixed step boundaries) to stabilize learning while allowing the target policy to improve. (iii) *Comparable budgets.* We report results under matched token budgets; if discounted training yields fewer reasoning tokens per generation, we increase generations to equalize total tokens seen before comparing accuracy. We now elaborate on each component.

## 4.1 EXTRINSIC VERSUS INTRINSIC REWARD

Extrinsic reward comes from the environment, whereas intrinsic reward is assigned by the learner to its own experience, usually to speed up learning or exploration (Singh et al., 2010; Barto, 2012; Linke et al., 2020). The goal of maximizing correctness is extrinsic, since it comes from the environment. By contrast, formatting rewards that encourage the learner to emit correctly structured reasoning and answer tags are intrinsic: they help the agent structure its reasoning and format the answer in a way that satisfies the verifier. Only the correctness reward is necessary to learn an optimal policy, but intrinsic rewards can guide the learner toward behaviors beneficial for learning. Since we care about learning Blackwell optimal policies, we discount only the extrinsic correctness reward and leave intrinsic formatting rewards undiscounted. Popular frameworks that allow discounting, such as ByteDance's Volcano Engine Reinforcement Learning for LLMs library (Sheng et al., 2025), discount both extrinsic and intrinsic rewards.

## 4.2 KL REGULARIZATION

Discounting strongly nudges the model to shorten its answers. If the policy moves too fast, it can *collapse*: it learns to stop early and forgets how to reason. We add a KL penalty to a *moving* reference policy to keep updates small—like a trust region—so the objective changes gradually. The reference policy is not fixed: we periodically refresh it to the current policy so the anchor follows progress without allowing a single large drift. More specifically, every $u$ training steps we perform

$$\pi_{\text{ref}} \leftarrow \texttt{stop\_grad}(\pi),$$

the details of which can be found in Gorbatovski et al. (2025) or in the TRL library (von Werra et al., 2020).

## 4.3 WHAT TO DISCOUNT

Discounting is applied only to reasoning (thinking) tokens:

$$K(\tau) = \sum_{t=1}^{|\tau|} m_t, \qquad m_t = \mathbb{I}\{\text{token } t \text{ lies in the reasoning span}\}.$$

In our experiments, we delineate the reasoning spans using explicit tags injected by prompting (e.g., `<reasoning> ··· </reasoning>`). Tokens required for prompt adherence, formatting and

the final answer segment have $m_t = 0$ and thus are not discounted. Empirically, discounting the entire response slightly hurt accuracy (about a $0.5\%$–$1.0\%$ drop on GSM8K): the model would occasionally drop formatting tags required by the verifier or respond with an answer that was too short (e.g., dropping zeros from long integers).

## 4.4 COMPARABLE TOKENS

Discounted policies produce shorter traces, so for the same number of epochs (or passes over prompts) they experience fewer transitions/samples than undiscounted policies. This can make discounted methods look worse simply because they saw less data, not because the objective is inferior. To keep comparisons fair, whenever this discrepancy mattered during training we adjusted the number of generations: either increasing generations for the discounted method or, when more sensible, decreasing generations for the undiscounted method so that the total samples/tokens observed were comparable.

In some settings, the discounted method still matched the undiscounted baseline despite seeing fewer samples—an informative robustness result. In others, we ensured sample counts were comparable to make a fair judgment.

**Practical notes.** (i) *Choosing $\gamma$.* In light of the Blackwell analysis, we select $\gamma$ as far from 1 as possible while preserving undiscounted training accuracy[2]. This can be accomplished via a simple bisection search, adjusting $\gamma$ until accuracy matches (or begins to dip below) the undiscounted training accuracy. (ii) *Updating the reference policy.* We choose the update frequency via ablations—namely, we find the best update frequency and $\beta$ that maximize the undiscounted model's accuracy and apply the same values to the discounted methods. (iii) *No algorithmic change required.* Any policy optimization algorithm—e.g., REINFORCE (Williams, 1992) and variants (such as RE-INFORCE Leave One Out (Ahmadian et al., 2024))—can be used with Equation (2); our contribution is training with the discounted return in Equation (1) together with the masking and budgeting rules above. In what follows, we employ GRPO as our policy optimization method.

## 5 NUMERICAL EXPERIMENTS

We empirically validate our theoretical prediction that discounting incentivizes efficient reasoning in large language models. Recall from Theorem 3.6 that in deterministic verifier MDPs, a Blackwell optimal policy prioritizes correctness and, among equally correct strategies, minimizes expected trajectory length. Our experiments test whether this pattern appears in practice when post training language models using GRPO.

**Setup.** We finetune and evaluate four instruction tuned models: Qwen2.5 7B-Instruct and Qwen2.5 14B-Instruct (Yang et al., 2025), Llama 3 8B-Instruct (Grattafiori et al., 2024) and Phi-4 (Abdin et al., 2024), post trained via GRPO with and without discounting. The undiscounted case ($\gamma = 1$) optimizes correctness only, whereas $\gamma < 1$ additionally rewards shorter successful trajectories. We evaluate on grade school math (GSM8K) (Cobbe et al., 2021) and MATH (Hendrycks et al., 2021) for Qwen2.5 7B and Llama 3 8B. We then train the larger Qwen2.5 14B and Phi-4 models on a subset of the DeepScaleR math dataset (Luo et al., 2025) and evaluate on AMC 2023, AIME 2025, MINERVA (Lewkowycz et al., 2022) and OLYMPIAD (He et al., 2024) to test generality. We report Pass@1 and mean response length. Pass@1 is the fraction of problems for which the first generated solution (one sample per prompt) is judged correct by the verifier. In our setting, the average Pass@1 is the accuracy.

**Implementation and benchmarking.** We use Hugging Face TRL for GRPO fine tuning and vLLM (Kwon et al., 2023) for inference. At inference, we use greedy decoding (temperature $\nu = 0$), consistent with Theorem A.12. We select Qwen2.5 7B-Instruct and Llama 3 8B-Instruct as established baselines for sanity checking our implementation and verify that our reimplementations meet or exceed published numbers on GSM8K and MATH. For Qwen2.5 7B-Instruct we compare against

---

[2]In our empirical setup, we first tune the hyperparameters to maximize the performance of the undiscounted method, then apply discounting with these hyperparameters.

| Dataset | Model | Undisc. Pass@1 | Undisc. Len | Disc. Pass@1 | Disc. Len |
|---------|-------|----------------|-------------|--------------|-----------|
| GSM8K | Qwen2.5 7B-Instruct | 91.06 | 217.60 | 91.07 | 170.08 |
|  | Llama 3 8B-Instruct | 80.87 | 125.43 | 81.07 | 108.67 |
| MATH | Qwen2.5 7B-Instruct | 64.80 | 491.32 | 64.55 | 384.96 |
|  | Llama 3 8B-Instruct | 24.48 | 328.43 | 24.75 | 257.73 |

Table 1: GSM8K and MATH: Pass@1 and mean response length (tokens) for discounted vs. undiscounted GRPO. Averaged over 3 training seeds and 10 evaluation seeds per model; evaluation seeds are fixed across methods for paired comparisons.

VERL's official baselines; for Llama 3 8B-Instruct we follow Roux et al. (2025). Minor differences may arise because we average over multiple training and evaluation seeds, whereas some prior reports use single seed estimates. For GSM8K we limit completion length to 786 tokens; for MATH to 2048 tokens; and for DeepScaleR to 4096 tokens.

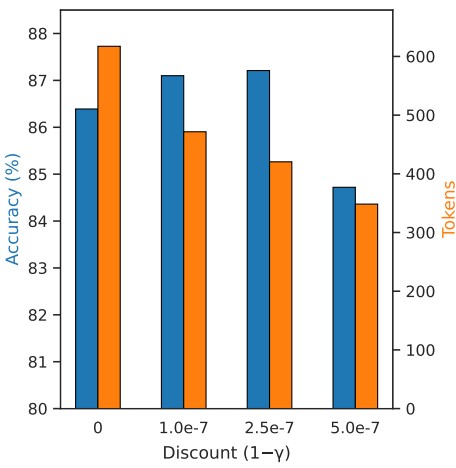

Figure 2: GSM8K accuracy (blue, left) and tokens (orange, right) vs. discount $(1 - \gamma)$.

**Variance control and reporting.** To obtain stable estimates, we repeat each training run with 3 random training seeds and, for each trained model, evaluate with 10 independent sampling seeds on GSM8K and MATH; we report averages over $3 \times 10$ runs per condition and fix evaluation seeds across methods for paired comparisons. For AMC 2023, AIME 2025, MINERVA and OLYMPIAD, we average over five evaluation seeds per model. This matters because RL style post training and decoding introduce variance (Patterson et al., 2024; He & Lab, 2025) and single seed reporting can be misleading for both Pass@1 and length statistics. When sweeping $\gamma$, we select and report a single discounted configuration per model/dataset using the following criterion: among all discounted settings whose training Pass@1 matches or exceeds that of the undiscounted run, we choose the one with the shortest mean response length. All tabled metrics are then computed on the evaluation seeds for the selected configuration.

**Main results.** Tables 1 and 2 show that, on average over seeds, discounted models match the accuracy of undiscounted ones while producing shorter responses. For example, on GSM8K, discounting reduces mean response length by 22% for Qwen2.5 7B-Instruct and by 13% for Llama 3 8B-Instruct with an insignificant change in Pass@1. This aligns with Theorem 3.6, which predicts shortest path behavior at fixed success probability. The trend holds for the larger models evaluated on datasets distinct from their training set. Specifically, the DeepScaleR math dataset does not contain problems from OLYMPIAD, MINERVA, or AIME 2025; however, it does include problems from AMC prior to 2023. Across architectures and datasets, we consistently observe that discounting enforces length minimization subject to maintaining accuracy.

**Effect of the discount factor.** We run additional experiments with Qwen3 1.7B (Yang et al., 2025) on GSM8K to examine performance as a function of $\gamma$. For these runs, we increase the completion length limit to 1536 because outputs were frequently clipped for being too long. As shown in Figure 2, varying $\gamma$ confirms the predicted tradeoff: smaller $\gamma$ reliably shortens responses but can reduce accuracy. Theory explains this: for $\gamma$ close to 1, policies first maximize correctness; overly aggressive discounting shifts probability toward shorter trajectories even when that harms success.

| Model | Dataset | Undisc. Pass@1 | Undisc. Len | Disc. Pass@1 | Disc. Len |
|---|---|---|---|---|---|
| Phi-4 | AMC 2023 | 51.00 | 1134.30 | 61.00 | 716.29 |
| | AIME 2025 | 14.00 | 1263.87 | 19.33 | 800.09 |
| | MINERVA | 28.46 | 553.74 | 29.85 | 318.10 |
| | OLYMPIAD | 36.91 | 1059.92 | 35.67 | 707.64 |
| Qwen2.5 14B-Instruct | AMC 2023 | 50.00 | 737.47 | 59.50 | 582.31 |
| | AIME 2025 | 10.00 | 891.43 | 10.67 | 699.56 |
| | MINERVA | 27.21 | 522.14 | 27.43 | 437.31 |
| | OLYMPIAD | 35.13 | 797.57 | 34.76 | 684.02 |

Table 2: Pass@1 and mean response length (tokens) for undiscounted vs. discounted GRPO. Averages over 5 evaluation seeds per model.

## 6 CONCLUSIONS AND FUTURE WORK

We studied efficient reasoning in verifier based MDPs through the lens of Blackwell optimality (Blackwell, 1962; Grand-Clément & Petrik, 2023). Within restricted policy classes, we showed that for $\gamma$ sufficiently close to 1 there exists a Blackwell optimal policy that maximizes undiscounted success and, among accuracy maximizers, minimizes expected trajectory length. For softmax training with greedy deployment, the induced deterministic deployment class is finite and admits a bounded Blackwell discount factor; we provide an explicit upper bound on how close to 1 the discount must be. Guided by this theory, we proposed a practical recipe: discount only the environment reward as a function of reasoning tokens, keep intrinsic formatting rewards undiscounted, add KL regularization to a moving reference policy (Peters et al., 2010) and ensure comparable token budgets. Empirically, discounted GRPO matches Pass@1 accuracy while substantially shortening responses across math benchmarks. Our theoretical results extend to methods that introduce small per token penalties in finite horizon MDPs with binary rewards (verifiers) (Bertsekas, 2012), suggesting that several length penalty methods (Arora & Zanette, 2025; Su & Cardie, 2025; Xiang et al., 2025) recover the same accuracy then length ordering in the near undiscounted regime when properly implemented. This further sheds light on adapting to the inherent token complexity of a given question (Lee et al., 2025): choosing $\gamma$ within the Blackwell region steers the learner toward the shortest successful trajectories allowed by the class without sacrificing accuracy. Some of our empirical results suggest that discounted methods can achieve higher accuracy with shorter reasoning traces. An interesting avenue for future work is to investigate whether shorter, more compressed reasoning improves generalization on reasoning tasks. As argued in Hutter (2007), compression (or prediction) is linked to improved generalization; whether this extends to compressed reasoning traces remains open. Another direction is to study whether methods that promote longer reasoning (Liu et al., 2025b) can be combined with methods that shorten reasoning: longer reasoning promotes path finding, while shorter reasoning promotes path compression. A pipeline that first uses longer traces to discover strategies and then compresses them (akin to distillation (Hinton et al., 2015)) may yield stronger reasoning policies.

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

## A    OMITTED PROOFS

**Proposition A.1.** *Fix $p \in (0, 1)$ and $0 < q_1 < q_2 < 1$, and consider the MDP in Figure 1 with horizon $H \geq 4$ and deterministic initial state $s_0$. Let the restricted policy class be $\Pi = \{\pi_1, \pi_2, \pi_3\}$, where at $s_0$: for $i \in \{1, 2\}$, $\pi_i$ selects $a_3$ with probability $q_i$ and $a_2$ with probability $1 - q_i$, and $\pi_3$ selects $a_1$ with probability $p$ and $a_4$ with probability $1 - p$. Let $\tau(\pi)$ denote the time step at which $a_{\text{end}}$ is taken under policy $\pi$. Then*

$$J_1(\pi_1) = J_1(\pi_2) = 1, \quad \mathbb{E}[\tau(\pi_i)] = 3 + q_i \ (i = 1, 2), \quad J_1(\pi_3) = p, \quad \tau(\pi_3) = 2.$$

*For all $\gamma \in [0, 1)$,*

$$J_\gamma(\pi_i) = (1 - q_i)\gamma^2 + q_i\gamma^3 \quad (i = 1, 2), \qquad J_\gamma(\pi_3) = p\gamma.$$

*Thus there exists a threshold $\gamma' \in (p, 1)$ such that for every $\gamma > \gamma'$, $\pi_1$ is both an optimal policy in $\Pi$ and a shortest path policy.*

*Proof.* Under $\pi_i$ for $i \in \{1, 2\}$, the process terminates successfully with reward 1 at time $\tau = 3$ if $a_2$ is chosen (probability $1 - q_i$) and at time $\tau = 4$ if $a_3$ is chosen (probability $q_i$). Hence $J_1(\pi_i) = 1$, $\mathbb{E}[\tau(\pi_i)] = 3(1 - q_i) + 4q_i = 3 + q_i$, and

$$J_\gamma(\pi_i) = (1 - q_i)\gamma^2 + q_i\gamma^3.$$

Under $\pi_3$, $a_{\text{end}}$ is taken at $\tau(\pi_3) = 2$ and yields reward 1 iff $a_1$ was chosen at $t = 1$, which occurs with probability $p$, hence $J_\gamma(\pi_3) = p\gamma$ and $J_1(\pi_3) = p$.

For $\gamma < 1$,

$$J_\gamma(\pi_1) - J_\gamma(\pi_2) = \left[(1 - q_1)\gamma^2 + q_1\gamma^3\right] - \left[(1 - q_2)\gamma^2 + q_2\gamma^3\right] = (q_2 - q_1)\gamma^2(1 - \gamma) > 0,$$

so discounting always prefers $\pi_1$ over $\pi_2$. Now consider

$$\phi(\gamma) := J_\gamma(\pi_1) - J_\gamma(\pi_3) = (1 - q_1)\gamma^2 + q_1\gamma^3 - p\gamma = \gamma\big((1 - q_1)\gamma + q_1\gamma^2 - p\big).$$

Let $f(\gamma) := (1 - q_1)\gamma + q_1\gamma^2 - p$. Then

$$f'(\gamma) = (1 - q_1) + 2q_1\gamma \geq 1 - q_1 > 0,$$

so $f$ is strictly increasing on $[0, 1]$. Moreover,

$$f(p) = (1 - q_1)p + q_1p^2 - p = -q_1p(1 - p) < 0, \qquad f(1) = 1 - p > 0,$$

so there exists a unique $\gamma_{\text{th}} \in (p, 1)$ with $f(\gamma_{\text{th}}) = 0$, i.e. $\phi(\gamma_{\text{th}}) = 0$. For $\gamma > \gamma_{\text{th}}$ we have $f(\gamma) > 0$ and hence $J_\gamma(\pi_1) > J_\gamma(\pi_3)$, while $J_\gamma(\pi_1) > J_\gamma(\pi_2)$ holds for all $\gamma < 1$. Thus for every $\gamma > \gamma_{\text{th}}$, $\pi_1$ is $\gamma$-optimal in $\Pi$; since $J_1(\pi_1) = J_1(\pi_2) = 1 > p = J_1(\pi_3)$ and $\mathbb{E}[\tau(\pi_1)] = 3 + q_1 < 3 + q_2 = \mathbb{E}[\tau(\pi_2)]$, $\pi_1$ is also a shortest path policy among accuracy maximizers. $\square$

We adapt classical Blackwell arguments (Zwick & Paterson, 1996; Puterman, 2014; Grand-Clément & Petrik, 2023) to the case where the admissible class is restricted. Throughout this section we assume finite horizon $H < \infty$, finite state and action sets and bounded rewards. For a policy $\pi$ and $\gamma \in [0, 1)$, define the (discounted) value

$$v_\gamma^\pi(s) := \mathbb{E}_{\pi,s}\Big[\sum_{t=1}^H \gamma^{t-1} R_t\Big], \qquad J_\gamma(\pi) := \int v_\gamma^\pi(s)\,\mu(ds).$$

We first handle a finite admissible class and then specialize to greedy deployment policies induced by a softmax training class.

**Assumption A.2** (Finite policy class). The admissible class $\Pi$ is finite: $|\Pi| < \infty$.

**Definition A.3.** Given $\gamma \in [0, 1)$, a policy $\pi \in \Pi$ is $\gamma$ discount optimal if $J_\gamma(\pi) \geq J_\gamma(\pi')$ for all $\pi' \in \Pi$. We call $\Pi_\gamma^\star \subset \Pi$ the set of $\gamma$ discount optimal policies.

**Definition A.4.** A policy $\pi$ is Blackwell optimal if there exists a $\gamma \in [0, 1)$ such that $\pi \in \Pi_{\gamma'}^\star$ for all $\gamma' \in [\gamma, 1)$. We call $\Pi_{\text{bw}}^\star$ the set of Blackwell optimal policies.

**Lemma A.5.** *For any $\pi, \pi' \in \Pi$, the difference $\Delta_{\pi,\pi'}(\gamma) := J_\gamma(\pi) - J_\gamma(\pi')$ is a polynomial in $\gamma$ of degree at most $H - 1$. Consequently it has finitely many roots in $[0, 1)$ unless it is identically zero.*

*Proof.* Linearity of expectation yields $J_\gamma(\pi) = \sum_{t=1}^{H} \gamma^{t-1} c_t(\pi)$ with $c_t(\pi) := \mathbb{E}_{\pi,\mu}[R_t]$, which is independent of $\gamma$. Subtracting $J_\gamma(\pi')$ and applying the fundamental theorem of algebra to $\sum_{t=1}^{H} \gamma^{t-1}(c_t(\pi) - c_t(\pi'))$ yields the result. $\qquad\square$

**Theorem A.6.** *Under Assumption A.2, there exists $\gamma' \in [0, 1)$ and a nonempty set $\Pi_{\mathrm{bw}}^\star \subseteq \Pi$ such that for all $\gamma \in (\gamma', 1)$,*
$$\underset{\pi \in \Pi}{\arg\max}\, J_\gamma(\pi) = \Pi_{\mathrm{bw}}^\star.$$

*Proof.* By Lemma A.5, each pairwise difference $\Delta_{\pi,\pi'}(\gamma)$ is a polynomial of degree at most $H - 1$. For each ordered pair $(\pi, \pi')$ with $\Delta_{\pi,\pi'} \not\equiv 0$, let $Z_{\pi,\pi'} = \{\gamma \in [0, 1) : \Delta_{\pi,\pi'}(\gamma) = 0\}$, which is finite. Define $\Gamma = \bigcup_{(\pi,\pi'): \Delta_{\pi,\pi'} \not\equiv 0} Z_{\pi,\pi'}$, which is finite. Set $\gamma' = \max \Gamma$ (or 0 if $\Gamma = \emptyset$). For any $\gamma \in (\gamma', 1)$ and any pair $\pi, \pi'$, either $\Delta_{\pi,\pi'} \equiv 0$ or it has no zeros in $(\gamma', 1)$, hence it has constant sign on that interval. Therefore all pairwise comparisons between $J_\gamma(\pi)$ and $J_\gamma(\pi')$ are fixed on $(\gamma', 1)$. It follows that $\Pi_\gamma^\star$ is constant on $(\gamma', 1)$; denote this common set by $\Pi_{\mathrm{bw}}^\star$. Nonemptiness follows from finiteness of $\Pi$. $\qquad\square$

**Definition A.7.** The Blackwell discount factor is
$$\gamma_{\mathrm{bw}} := \inf \left\{ \gamma \in [0, 1) : \ \Pi_{\gamma'}^\star = \Pi_{\mathrm{bw}}^\star \ \forall \gamma' \in (\gamma, 1) \right\},$$

where $\Pi_\gamma^\star = \arg\max_{\pi \in \Pi} J_\gamma(\pi)$.

**Lemma A.8.** *Under Assumption A.2, the Blackwell factor $\gamma_{\mathrm{bw}}$ exists and satisfies $\gamma_{\mathrm{bw}} < 1$.*

*Proof.* Theorem A.6 ensures that $\Pi_\gamma^\star$ is constant for all $\gamma$ sufficiently close to 1, so the infimum in Definition A.7 is well defined and strictly less than 1. $\qquad\square$

### A.1 Softmax training, greedy deployment

Let $\Pi_s$ be the (possibly infinite) class of softmax policies used during training. We use the standard *time-augmented, stationary, infinite-horizon* representation of the finite-horizon problem with horizon $H$. Define the augmented state space:
$$\tilde{\mathcal{S}} = \{(s, t) : s \in \mathcal{S}, \ t \in \{1, \dots, H\}\} \cup \{\texttt{absorb}\},$$

and the stationary transition kernel $\tilde{P}$ and rewards $\tilde{r}$ by
$$\tilde{P}\big((s', t{+}1) \mid (s, t), a\big) = P(s' \mid s, a) \quad (t < H),$$
$$\tilde{P}(\texttt{absorb} \mid (s, H), a) = 1, \qquad \tilde{P}(\texttt{absorb} \mid \texttt{absorb}, a) = 1,$$
$$\tilde{r}\big((s, t), a\big) = r(s, a) \quad (t \le H), \qquad \tilde{r}(\texttt{absorb}, a) = 0.$$

The initial distribution on augmented states is $\tilde{\mu}$ with $\tilde{\mu}((s, 1)) = \mu(s)$ and zero elsewhere. A (possibly nonstationary) finite-horizon policy $\pi = (\pi_t)_{t=1}^{H}$ induces the stationary policy
$$\tilde{\pi}(a \mid (s, t)) = \pi_t(a \mid s) \quad (t \le H), \qquad \tilde{\pi}(\cdot \mid \texttt{absorb}) \text{ arbitrary}.$$

We fix a deterministic tie breaking rule on $\mathcal{A}$ and define the greedification map on augmented states
$$\mathrm{Greed}(\pi, (s, t)) \in \arg\max_{a \in \mathcal{A}} \pi_t(a \mid s) \qquad \forall (s, t) \in \mathcal{S} \times \{1, \dots, H\}.$$

The deployment class is the image $\Sigma := \{\mathrm{Greed}(\pi, \cdot) : \pi \in \Pi_s\}$, a subset of the *deterministic stationary* policies on the augmented MDP $\tilde{M} = (\tilde{\mathcal{S}}, \mathcal{A}, \tilde{P}, \tilde{r}, \tilde{\mu})$. Each $\sigma \in \Sigma$ corresponds one-to-one to a deterministic nonstationary policy on the original depth-$H$ decision tree.

**Lemma A.9.** *The set $\Sigma$ is finite. In particular, if $N_{\mathrm{nodes}}$ is the number of reachable decision nodes up to depth $H$ in the original tree, then $|\Sigma| \le |\mathcal{A}|^{N_{\mathrm{nodes}}}$.*

*Proof.* Finite states and finite horizon imply a finite reachable decision tree. A greedy policy assigns exactly one action to each reachable node (equivalently, to each reachable augmented state $(s, t)$ with $t \leq H$), so the number of labelings is at most $|\mathcal{A}|^{N_{\text{nodes}}}$. $\qed$

For any policy class $\Pi$, let $\gamma_{\text{bw}}(\Pi)$ denote the Blackwell discount factor given that class in the (augmented) stationary MDP. By Theorem A.6 with $\Pi \leftarrow \Sigma$, we obtain:

**Corollary A.10.** *There exists* $\gamma_{\text{bw}}(\Sigma) < 1$ *and a nonempty set* $\Sigma_{\text{bw}}^\star \subseteq \Sigma$ *such that* $\arg\max_{\pi \in \Sigma} J_\gamma(\pi) = \Sigma_{\text{bw}}^\star$ *for all* $\gamma \in (\gamma_{\text{bw}}(\Sigma), 1)$.

**Setup** For a stationary deterministic policy $\pi$ on $(\tilde{\mathcal{S}}, \mathcal{A})$, let $P_\pi$ and $r_\pi$ be the induced transition matrix and reward vector on $\tilde{\mathcal{S}}$. Define the $\mu$-weighted discounted return through the augmented value equation

$$v_\gamma^\pi = r_\pi + \gamma P_\pi v_\gamma^\pi, \qquad J_\gamma(\pi) = \tilde{\mu}^\top v_\gamma^\pi,$$

so that, for any finite-horizon policy and its image under time-augmentation, the objectives coincide: $J_\gamma(\text{finite-horizon } \pi) = J_\gamma(\text{stationary } \tilde{\pi})$ for all $\gamma \in [0, 1)$. For a polynomial $p(X) = \sum_{k=0}^N a_k X^k$, write the coefficient extractor $[X^k]p = a_k$.

For $\pi, \pi'$ in the admissible class $\Pi$ (we will take $\Pi = \Sigma$), set

$$\gamma_\mu(\pi, \pi') := \max\left\{\gamma \in [0, 1) : \tilde{\mu}^\top\left(v_\gamma^\pi - v_\gamma^{\pi'}\right) = 0\right\},$$

with the convention $\gamma_\mu(\pi, \pi') = 0$ if the above set is empty or if $J_\gamma(\pi) - J_\gamma(\pi') \equiv 0$ on $[0, 1)$. We aim to upper bound

$$\bar{\gamma} = \max_{\pi, \pi' \in \Pi} \gamma_\mu(\pi, \pi').$$

(If one restricts to a subclass $\Pi' \subseteq \Pi$, replace $\Pi$ by $\Pi'$ everywhere; the bound below only becomes easier.)

**Assumption A.11.** There exists $m \in \mathbb{N}$ such that for any $(s, a, s') \in \mathcal{S} \times \mathcal{A} \times \mathcal{S}$,

$$P(s'|s, a) = \frac{n(s, a, s')}{m}$$

with $n(s, a, s') \in \mathbb{Z}_{\geq 0}$, $n(s, a, s') \leq m$ and

$$r(s, a) = \frac{w(s, a)}{m}$$

with $w(s, a) \in \mathbb{Z}$ and $|w(s, a)| \leq r_\infty$.

The augmented kernel $\tilde{P}$ and rewards $\tilde{r}$ inherit this structure. Let $D_{\tilde{\mu}} = \min\{t \in \mathbb{N}_{>0} : t\tilde{\mu} \in \mathbb{Z}^{\tilde{\mathcal{S}}}\}$ be the least positive integer such that $t\tilde{\mu}$ is integer-valued.

**Theorem A.12.** *Under Assumption A.11, for any rational* $\mu \in \Delta(\mathcal{S})$ *define*

$$N = 2|\tilde{\mathcal{S}}| - 1, \qquad L_\mu = 2 D_{\tilde{\mu}} |\tilde{\mathcal{S}}| r_\infty m^{2|\tilde{\mathcal{S}}|} 4^{|\tilde{\mathcal{S}}|}, \qquad \eta_\mu = \frac{1}{2 N^{N/2+2} (L_\mu + 1)^N}.$$

*Then, with* $\bar{\gamma} = \max_{\pi, \pi' \in \Sigma} \gamma_\mu(\pi, \pi')$,

$$\bar{\gamma} < 1 - \eta_\mu.$$

*Proof.* All objects are on the augmented state space $\tilde{\mathcal{S}}$. By Cramer's rule (Lemma A.1 of Grand-Clément & Petrik (2023)), for any deterministic $\pi$ we have

$$v_\gamma^\pi(s) = \frac{n(\gamma, s, \pi)}{d(\gamma, \pi)}, \qquad d(\gamma, \pi) = \det(I - \gamma P_\pi), \qquad n(\gamma, s, \pi) = \det\left(M(\gamma, s, \pi)\right),$$

where $M(\gamma, s, \pi)$ is formed by replacing the $s$-th column of $I - \gamma P_\pi$ by $r_\pi$. Writing $\bar{n}(\gamma, \pi) := \sum_{s \in \tilde{\mathcal{S}}} \tilde{\mu}(s) n(\gamma, s, \pi)$, we get

$$J_\gamma(\pi) = \frac{\bar{n}(\gamma, \pi)}{d(\gamma, \pi)}, \qquad J_\gamma(\pi) - J_\gamma(\pi') = \frac{p(\gamma)}{d(\gamma, \pi)d(\gamma, \pi')},$$

with

$$p(\gamma) := \bar{n}(\gamma, \pi)\, d(\gamma, \pi') - \bar{n}(\gamma, \pi')\, d(\gamma, \pi).$$

By Lemma A.2 of Grand-Clément & Petrik (2023), $d(\gamma, \pi) > 0$ on $[0,1)$ and by Lemma A.3, $p(1) = 0$. Since $\deg \bar{n} \leq |\tilde{\mathcal{S}}| - 1$ and $\deg d \leq |\tilde{\mathcal{S}}|$, we have $\deg p \leq N := 2|\tilde{\mathcal{S}}| - 1$.

By Proposition A.6 of Grand-Clément & Petrik (2023), $m^{|\tilde{\mathcal{S}}|} n(\cdot, s, \pi)$ has integer coefficients and

$$\sum_{k=0}^{N} \left| [X^k]\big(m^{|\tilde{\mathcal{S}}|} n(\cdot, s, \pi)\big)\right| \; \leq \; |\tilde{\mathcal{S}}|\, r_\infty\, m^{|\tilde{\mathcal{S}}|}\, 2^{|\tilde{\mathcal{S}}|}.$$

Thus $m^{|\tilde{\mathcal{S}}|} D_{\tilde{\mu}}\, \bar{n}(\cdot, \pi) = \sum_s \big(D_{\tilde{\mu}}\tilde{\mu}(s)\big)\, m^{|\tilde{\mathcal{S}}|} n(\cdot, s, \pi)$ has integer coefficients and coefficient-sum at most $D_{\tilde{\mu}}\, |\tilde{\mathcal{S}}|\, r_\infty\, m^{|\tilde{\mathcal{S}}|}\, 2^{|\tilde{\mathcal{S}}|}$. By Proposition A.5 of Grand-Clément & Petrik (2023), $m^{|\tilde{\mathcal{S}}|} d(\cdot, \pi)$ has integer coefficients and

$$\sum_{k=0}^{N} \left| [X^k]\big(m^{|\tilde{\mathcal{S}}|} d(\cdot, \pi)\big)\right| \; \leq \; m^{|\tilde{\mathcal{S}}|}\, 2^{|\tilde{\mathcal{S}}|}.$$

Applying Proposition A.7 of Grand-Clément & Petrik (2023) to the two products defining $p(\gamma)$ and summing, we obtain that

$$\tilde{p}(\gamma) := m^{2|\tilde{\mathcal{S}}|} D_{\tilde{\mu}}\, p(\gamma)$$

has integer coefficients and

$$\sum_{k=0}^{N} \big| [X^k]\tilde{p} \big| \; \leq \; 2\left(D_{\tilde{\mu}}\, |\tilde{\mathcal{S}}|\, r_\infty\, m^{|\tilde{\mathcal{S}}|}\, 2^{|\tilde{\mathcal{S}}|}\right) \cdot \left(m^{|\tilde{\mathcal{S}}|} 2^{|\tilde{\mathcal{S}}|}\right) \; = \; L_\mu.$$

The degree of $\tilde{p}$ is at most $N$ and $\tilde{p}$ shares roots with $p$. By Theorem A.8 in Grand-Clément & Petrik (2023), any two distinct roots of an integer-coefficient degree-$N$ polynomial with coefficient-sum $\leq L_\mu$ are at distance at least $\eta_\mu = \left[2\, N^{N/2+2}\, (L_\mu + 1)^N\right]^{-1}$. If the set in the definition of $\gamma_\mu(\pi, \pi')$ is empty, then $\gamma_\mu(\pi, \pi') = 0$ and the claim holds trivially. Otherwise, 1 and $\gamma_\mu(\pi, \pi') \in [0,1)$ are distinct roots, hence $\gamma_\mu(\pi, \pi') \leq 1 - \eta_\mu$. Maximizing over $\pi, \pi' \in \Sigma$ gives $\bar{\gamma} < 1 - \eta_\mu$. □

**Corollary A.13.** *For any $\Sigma' \subseteq \Sigma$, the same bound holds with $\bar{\gamma}$ replaced by $\max_{\pi, \pi' \in \Sigma'} \gamma_\mu(\pi, \pi')$.*

**Assumption A.14.** There exists a termination action $a_{\text{term}} \in \mathcal{A}$ (e.g., eos), an absorbing state $s_{\text{abs}} \in \mathcal{S}$, and a goal set $G \subseteq \mathcal{S}$ such that for all $s \in \mathcal{S}$:

1. $r(s, a) = 0$ for all $a \neq a_{\text{term}}$;

2. taking $a_{\text{term}}$ transitions to the absorbing state, i.e. $P(s_{\text{abs}} \mid s, a_{\text{term}}) = 1$;

3. the terminal reward is deterministic and binary, $r(s, a_{\text{term}}) = \mathbb{I}\{s \in G\} \in \{0, 1\}$. Moreover, the absorbing state yields no further reward and transitions to itself: for all $a \in \mathcal{A}$,

$$r(s_{\text{abs}}, a) = 0, \qquad P(s_{\text{abs}} \mid s_{\text{abs}}, a) = 1.$$

Let $\tau \leq H$ be the (first) absorption time. Define the success probability and (conditional) successful-path length

$$p(\pi) = \mathbb{P}_{\pi, \mu}(\text{success within } H), \qquad L(\pi) = \mathbb{E}_{\pi, \mu}[\tau \mid \text{success}],$$

with the convention that $L(\pi)$ is only evaluated when $p(\pi) > 0$. Call $\pi$ a *shortest-path policy* if it maximizes $p(\pi)$ and, among all maximizers of $p$, minimizes $L(\pi)$. If $p_\star := \max_\pi p(\pi) = 0$, the shortest-path condition reduces to the first criterion.

**Lemma A.15.** *Let $\varepsilon = 1 - \gamma$. For every policy $\pi$,*

$$J_\gamma(\pi) = \mathbb{E}_{\pi, \mu}\big[\gamma^{\tau - 1} \mathbf{1}\{\text{success}\}\big] = p(\pi)\Big(1 - \varepsilon\,(L(\pi) - 1)\Big) + R_\pi(\varepsilon),$$

*with remainder satisfying the uniform bound $|R_\pi(\varepsilon)| \leq C_H\, \varepsilon^2$, where $C_H := \frac{1}{2}(H-1)(H-2)$.*

*Proof.* Since the reward is 1 only upon successful termination at time $\tau$,

$$J_\gamma(\pi) = \mathbb{E}_{\pi,\mu}\big[\gamma^{\tau-1}\mathbf{1}\{\text{success}\}\big] = p(\pi)\,\mathbb{E}\big[(1-\varepsilon)^{\tau-1} \mid \text{success}\big],$$

where $\varepsilon = 1 - \gamma$. For any integer $n \in \{0,\dots,H-1\}$, Taylor's theorem gives that for some $\xi \in (0,\varepsilon)$,

$$(1-\varepsilon)^n = 1 - n\varepsilon + \tfrac{1}{2}n(n-1)(1-\xi)^{n-2}\varepsilon^2,$$

which, since $\xi \in [0,1)$, implies

$$\big|(1-\varepsilon)^n - (1-n\varepsilon)\big| \le \tfrac{1}{2}n(n-1)\varepsilon^2.$$

Setting $n = \tau - 1 \in \{0,\dots,H-1\}$ and conditioning on success yields

$$\mathbb{E}\big[(1-\varepsilon)^{\tau-1} \mid \text{success}\big] = 1 - \varepsilon\,(L(\pi)-1) + \mathbb{E}[\delta_{\tau-1}(\varepsilon) \mid \text{success}],$$

with $|\delta_{\tau-1}(\varepsilon)| \le \tfrac{1}{2}(\tau-1)(\tau-2)\varepsilon^2 \le C_H\varepsilon^2$. Define $R_\pi(\varepsilon) := p(\pi)\,\mathbb{E}[\delta_{\tau-1}(\varepsilon) \mid \text{success}]$ to conclude. $\qquad\square$

**Theorem A.16.** *In finite-horizon MDPs with a deterministic binary terminal verifier reward (Assumption A.14), every Blackwell optimal policy is a shortest path policy:*

$$\Pi^\star_{\mathrm{bw}} \subseteq \operatorname*{argmin}_{\pi \in \Pi_{\max p}} L(\pi), \quad \text{where} \quad \Pi_{\max p} = \operatorname*{argmax}_{\pi \in \Pi} p(\pi).$$

*Proof.* Let $\pi^\star \in \Pi^\star_{\mathrm{bw}}$. For any $\pi \in \Pi$ and $\varepsilon = 1 - \gamma$, Lemma A.15 gives

$$J_\gamma(\pi^\star) - J_\gamma(\pi) = \underbrace{p(\pi^\star) - p(\pi)}_{(A)} - \varepsilon\underbrace{\big(p(\pi^\star)(L(\pi^\star)-1) - p(\pi)(L(\pi)-1)\big)}_{(B)} + \underbrace{R_{\pi^\star}(\varepsilon) - R_\pi(\varepsilon)}_{(C)},$$

with $|(C)| \le 2C_H\varepsilon^2$. If $p(\pi) > p(\pi^\star)$, then for sufficiently small $\varepsilon > 0$ the RHS is negative, contradicting optimality of $\pi^\star$ for $\gamma$ arbitrarily close to 1. Hence $p(\pi^\star) \ge p(\pi)$ for all $\pi$, i.e., $\pi^\star \in \Pi_{\max p}$.

Now fix any $\pi \in \Pi_{\max p}$ so that $p(\pi) = p(\pi^\star) = p_\star$. If $L(\pi) < L(\pi^\star)$ then $(B) = p_\star(L(\pi^\star) - L(\pi)) > 0$ and for small enough $\varepsilon$ the negative term $-\varepsilon\,(B)$ dominates the $O(\varepsilon^2)$ remainder, again contradicting optimality. Therefore $L(\pi^\star) \le L(\pi)$ for all $\pi \in \Pi_{\max p}$.

The same argument applies with $\Pi$ replaced by any subclass (e.g., the finite deployment class $\Sigma$). $\qquad\square$

**Corollary A.17.** *In the time-augmented reasoning MDP, the Blackwell-optimal deployed policies satisfy*

$$\Sigma^\star_{\mathrm{bw}} = \arg\min_{\sigma \in \Sigma_{\max p}} L(\sigma), \qquad \Sigma_{\max p} := \arg\max_{\sigma \in \Sigma} p(\sigma).$$

*Equivalently, for $\gamma$ sufficiently close to 1, the $\gamma$-discounted optimal policies in $\Sigma$ are exactly the shortest successful-path policies.*

