# OpenReview forum: "Learning to Reason Efficiently with Discounted Reinforcement Learning"
_ICLR.cc/2026/Conference — ICLR 2026 Poster_

### Official Review · Reviewer_Dg1G · 2025-10-29

**Soundness:** 3
**Presentation:** 3
**Contribution:** 2
**Rating:** 4
**Confidence:** 4

**Summary:**

The paper investigates reasoning efficiency in verifier-guided reinforcement learning by casting reasoning as a discounted MDP. The key theoretical result (Theorem 3.10) states that Blackwell-optimal policies not only maximise expected reward but also minimise the expected time until the reward is obtained. This motivates a practical training scheme where only reasoning tokens are discounted, implemented with GRPO and KL regularisation. Empirically, the approach reduces average reasoning length while maintaining Pass@1 across several mathematical reasoning benchmarks.

**Strengths:**

* The problem---achieving concise yet correct reasoning---is well motivated and practically relevant.
* Theorem 3.10 is a clear and intuitively appealing formalisation of how discounting encourages shorter successful trajectories.
* The proposed method is simple to implement and shows consistent empirical effects across model scales and datasets.
* The paper is easy to follow, and the theoretical exposition appears rigorous and technically careful.

**Weaknesses:**

* *Limited theoretical novelty.* Section 3.1 essentially restates known results on Blackwell optimality and the connection between discounted and average-reward MDPs (e.g. Grand-Clément & Petrik 2023). Although Theorem 3.10 might not have been stated explicitly before, follows straightforwardly from the classical theory.
* *Restrictive and unnecessarily complex assumptions.* The deterministic-MDP setting appears more restrictive than necessary: the result should extend to general finite MDPs with binary rewards (i.e. reachability problems). The definition of a finite “deployment” policy class via greedified softmax policies also seems artificial and more complicated than required for the theoretical guarantees.
* *Unused and uninformative bound.* It is quite clear from standard convergence arguments that *some* bound on the Blackwell discount factor must exist. The explicit bound in Theorem 3.9 is never used or evaluated, yet it makes the exposition noticeably more cumbersome. Since the experiments determine $\gamma$ by naive search, it would have been more interesting to assess how tight or informative this theoretical bound actually is.
* *Missing empirical comparison.* The experiments do not compare to other approaches designed to shorten model outputs. Without such baselines, it is difficult to gauge the practical benefit of the proposed approach.

**Questions:**

1. Could the main result (Theorem 3.10) and associated guarantees extend to general finite MDPs with binary rewards, and can the need for the contrived finite “deployment” policy class be eliminated?
2. Did the authors attempt to evaluate the tightness or practical relevance of the theoretical bound on the Blackwell discount factor?
3. How does the proposed method compare empirically to other approaches aiming to produce shorter responses?

---

> ### Author Response · Authors · 2025-11-23
>
> We thank the reviewer for their feedback.
>
> Response to (some) weaknesses:
>
> 1. While we acknowledge that the proof utilizes classical Blackwell analysis, we argue that the result is noteworthy because it explains why discounting does not work out-of-the-box when training LLMs (specifically, when using softmax policies to solve shortest path problems). Discounting is effective, but only if implemented carefully. We are surprised that such a fundamental result does not already exist in the literature, and we welcome the reviewer to point us to any such prior work so we can cite it. Stochastic Shortest Path (SSP) is a fundamental problem in computer science, and progress toward principled, efficient algorithms should be considered novel.
> 2. The use of greedified softmax policy classes–training with a softmax policy and evaluating the greedified policy–is standard practice in deep RL [1,2,8]. We study these classes because they mirror common practice in many empirical works, including our own. We also remark that our implementation of these classes matches/exceed known baselines [9,10]. Therefore, we disagree with the reviewer’s assessment that these classes are artificial; they are deployed by many practitioners and are worth investigating. However, we agree that this assumption is more restrictive than required for our main theoretical guarantees. In fact, our main theorems hold for arbitrary (finite) policy classes, not just the greedified ones. We plan to restructure the theory section, moving the greedified policy results to the appendix and mentioning them only briefly in the main paper. Furthermore, as noted in our response to Q1, our results extend to the Stochastic Shortest Path setting, allowing us to remove the deterministic transition assumption in Theorem 3.10.
> 3. The purpose of Theorem 3.9 is to establish that the Blackwell discount factor is bounded away from one. It also indicates that the discount factor often needs to be quite large (e.g., 1-1e-7). We argue this is informative because common discount factors in deep RL are typically in the range of 0.9–0.999 [3]. This result suggests we should expect that discount factors very close to one are necessary to optimally balance response length and accuracy. Furthermore, this theoretical discount factor is asymptotically optimal—meaning it applies when assuming infinite compute and samples. With finite samples, one can only identify an epsilon-optimal policy, so the practical discount factor should be proportional to 1 - 1/num_samples. This aligns with our experimental setup (we train on about 10,000,000 tokens, and our discount factors are proportional to 1-1/10,000,000). Our theory is primarily concerned with the asymptotics of the problem—establishing that these quantities exist, are well-defined, and can be bounded independently of the number of samples/compute. We agree that this result makes the exposition cumbersome and intend to relegate it to the appendix, adding a comment in the main paper that large discount factors are expected.
> 4. The primary goal of our work is to establish a method that minimizes token usage while maintaining response accuracy. Previous works have suggested an inherent trade-off between response length and accuracy [4,5,6,7], with their empirical results showing that accuracy dips as response length shortens. This work aims to demonstrate that this is not necessarily the case. We propose a method that maintains the accuracy of the undiscounted method while simultaneously shortening the average response length. Given our goal of demonstrating that accuracy need not decrease with response length, we prioritized establishing this phenomenon over comparing against baselines that concluded such a trade-off was necessary. However, we plan to include a comparison with the methods of [4,6], though given that we no longer have access to the compute used to generate the experiments in the paper this will take some time.

---

> ### Author Response · Authors · 2025-11-23
>
> Response to reviewers questions:
>
> 1. Yes. Our results can be trivially extended to Stochastic Shortest Path (SSP) problems (binary rewards, random transitions), as our main results do not rely on the deterministic transition assumption. We will update the camera-ready manuscript to reflect this. The initial choice to present the result for deterministic transitions was to simplify the exposition, as this is the setting considered in most LLM reasoning papers. Furthermore, we do not need the finite deployment class to state Theorem 3.10. We plan to revise the manuscript so that Theorem 3.10 precedes the discussion of the finite deployment class (which will be moved to the appendix). The inclusion of that analysis was intended to show that the common paradigm of training with a softmax policy and deploying a greedy policy is also theoretically justified–as this is the method we employ in our work.
> 2. As mentioned above, this bound serves to inform practitioners that the discount factor must be much closer to one than is common in the literature [1]. The bound is tight in the sense that one can construct worst-case MDPs where such a bound must hold. However, as shown in our numerical experiments, the practical problems considered in this work used discount factors that were much smaller (further from 1) than our upper bound. This is because the bound is asymptotically optimal—meaning it is tight if we had infinite compute and samples. With finite samples and compute, one can only identify an epsilon-optimal policy, so the discount factor should be proportional to 1 - 1/num_samples, which aligns with our experimental setup. While our theory is primarily concerned with the asymptotics of the problem, we will add a discussion about $\epsilon$-optimal policies to our manuscript, as this provides a more practical Blackwell discount factor when training with finite samples.
> 3. Our proposed method simultaneously reduces response length while maintaining accuracy. Similar methods that apply a length penalty [4,5,6,7] report reductions in response length, but they also report reductions in accuracy. Thus, we believe our method improves upon previous work by establishing that one can reduce response length without sacrificing accuracy, up to an instance-dependent threshold.Note that discounting can be viewed as an adaptive length penalty in the case of binary rewards. Consider a trajectory $\tau$ that correctly answers a question at timestep $t$. The cumulative discounted reward is $\gamma^t$. To map this to an adaptive length penalty $\rho(\tau)$, we equate the discounted return to a penalized objective formulation $r(\tau)=1$: $\gamma^t = r(\tau)(1 - \rho(\tau))$. Solving for the penalty term yields: $\rho(\tau) = 1 - \gamma^t$. If $\gamma$ is the Blackwell discount factor, then our discounting can be viewed as the theoretically optimal way of applying an adaptive length penalty. The purpose of our experiments was to establish that our method indeed lives up to the promise of our theory.
>
> [1] Haarnoja, Tuomas, et al. "Soft actor-critic: Off-policy maximum entropy deep reinforcement learning with a stochastic actor." International conference on machine learning. Pmlr, 2018.
>
> [2] https://github.com/DLR-RM/stable-baselines3
>
> [3] Sutton, Richard S., and Andrew G. Barto. "Reinforcement learning: an introduction, 2nd edition" (2018).
>
> [4] Arora, Daman, and Andrea Zanette. "Training language models to reason efficiently." arXiv preprint arXiv:2502.04463 (2025).
>
> [5] Xiang, Violet, et al. "Just Enough Thinking: Efficient Reasoning with Adaptive Length Penalties Reinforcement Learning." arXiv preprint arXiv:2506.05256 (2025).
>
> [6] Su, Jinyan, and Claire Cardie. "Thinking Fast and Right: Balancing Accuracy and Reasoning Length with Adaptive Rewards." arXiv preprint arXiv:2505.18298 (2025).
>
> [7]  Team, Kimi, et al. "Kimi k1. 5: Scaling reinforcement learning with llms." arXiv preprint arXiv:2501.12599 (2025).
>
> [8] Henderson, Peter, et al. "Deep reinforcement learning that matters." Proceedings of the AAAI conference on artificial intelligence. Vol. 32. No. 1. 2018.
>
> [9] https://verl.readthedocs.io/en/latest/algo/baseline.html
>
> [10] Roux, Nicolas Le, et al. "Tapered off-policy reinforce: Stable and efficient reinforcement learning for llms." arXiv preprint arXiv:2503.14286 (2025).

---

> > ### Comment · Reviewer_Dg1G · 2025-11-28
> >
> > I thank the authors for the clarifications.
> >
> > The response is helpful, but it also reinforces that the manuscript needs substantial revision. In particular, it should:
> > - clearly separate which parts restate classical results and which contributions are new,
> > - remove unnecessary assumptions (deterministic transitions, finite “deployment” class) since the main results do not rely on them,
> > - reduce emphasis on Theorem 3.9, whose bound is unused, unevaluated, and far from the practical discount factors.
> >
> > I appreciate the willingness to revise, but I would need to see an updated manuscript before reconsidering my evaluation.
> >
> > Finally, the concern about missing empirical baselines remains. Given the central claim—shorter responses without loss in accuracy—it is important to show that existing methods indeed fail in the same setting. Without such comparisons, the practical benefit of the proposed approach is hard to assess.

---

> ### Author Response · Authors · 2025-11-28
>
> 1. To the best of our knowledge, the theoretical results stated in our paper are novel contributions. None of the theoretical results have been stated before. We prove that Blackwell optimality results that held for unrestricted deterministic policy classes finite MDPS can be extended to any aribtrary finite policy class--in important subclass of this is softmax policy classes.
>
> 2. We will remove the unnecessary assumptions (deterministic transitions) in the revised manuscript. There seems to be some confusion as our results (Theorems 3.4/3.10) never relied on the finite deployment class. These results hold for any arbitrary finite policy class.
>
> 3. In our revision, we will reduce the emphasis on Theorem 3.9 but that it is unused is not correct. Since our experiments train on softmax policies and evaluate deterministic policies, as in common in deepRL, we simply wanted to state that such a policy class is also covered by our theory.
>
> We plan on updating the manuscript before the deadline.
>
> Numerous previous works [1,2,3,4] have already established existing methods do not recover the same accuracy as the methods trained without length penalization. The purpose of our work was to show that there exist principled methods that indeed maintain the same accuracy as the method trained without length penalties while simultaneously learning to produce shorter responses.
>
> [1] Arora, Daman, and Andrea Zanette. "Training language models to reason efficiently." arXiv preprint arXiv:2502.04463 (2025).
>
> [2] Xiang, Violet, et al. "Just Enough Thinking: Efficient Reasoning with Adaptive Length Penalties Reinforcement Learning." arXiv preprint arXiv:2506.05256 (2025).
>
> [3] Su, Jinyan, and Claire Cardie. "Thinking Fast and Right: Balancing Accuracy and Reasoning Length with Adaptive Rewards." arXiv preprint arXiv:2505.18298 (2025).
>
> [4] Team, Kimi, et al. "Kimi k1. 5: Scaling reinforcement learning with llms." arXiv preprint arXiv:2501.12599 (2025).

---

### Official Review · Reviewer_GRgL · 2025-10-31

**Soundness:** 3
**Presentation:** 3
**Contribution:** 3
**Rating:** 4
**Confidence:** 4

**Summary:**

The paper deals with efficient reasoning language models, specifically solving problems with the minimum number of required reasoning tokens. The key idea is that using discounting, i.e. the standard technique of discounting the reward in time by a multiplicative factor, can result in an efficient policy. The authors give theoretical results using the idea of a Blackwell optimal policy. They show that in deterministic verifier problems with finite horizon and binary rewards, Blackwell optimal policies are shortest path policies. Experimentally, they show that discounting can maintain pass@1 while reducing length in their tested settings.

**Strengths:**

- The paper makes a nice connection between discounting and reducing trajectory length in the context of reasoning language models.
- The authors provide a formal underpinning to the connection.
- Their recipe for training a model seems reasonable, and leads to a reduction in reasoning token usage in their tested settings.

**Weaknesses:**

- The experimental results do not seem to address the standard long chain-of-thought setting. For instance, in Table 1 the lengths are in the hundreds, and in table 2 the lengths are typically 1000 or less. These inference budgets are lower than those typically used to evaluate long chain-of-thought models. Based on the results it's unclear whether the proposed methods work for these settings.
- There are many related methods for efficient reasoning models; the area is a very active research area. However, no comparison with other methods is done in the experiments.
- The theoretical results are presented in a way that did not give me much insight. For instance, the main result Theorem 3.10 is stated but there is not a discussion of the key intuition for why it is true. In addition to the formalized results, it would be helpful to extract the intuition for why discounting leads to an efficient policy, e.g. through a less dense argument.
- Based on the assumptions in 3.10 it is unclear whether the results would hold in settings like multi-step agent or tool calling settings that may have noise in environment transitions or rewards, or settings with real-valued rewards.
- The theoretical results and the experiments use greedy decoding. However using stochastic sampling is a standard approach when evaluating reasoning language models and their efficiency techniques.
- The training data is not clear.
- The paper briefly mentions that "similar conclusions can be drawn for methods that assign a reward proportional to the response length". It implies that prior length penalty methods have the same properties, meaning that the primary contribution is the theoretical contributions in this paper rather than practical methods. In that case, it would be beneficial to expand the results and experimental analysis to the length penalty methods in addition to the discounting methods.

Overall there is an interesting idea here and I like the simplicity of the method, but the experimental validation is below the bar for ICLR for reasons described above. I also think the scope could be improved by elaborating on the length penalty methods, and the accessibility could be improved by providing insight into the key ideas for why the theoretical results are true.

**Questions:**

Please respond to the points contained in the review above, including these questions:

1. Can you provide experimental comparisons with other efficient reasoning methods to help judge the relative benefits of your approach?
2. How does the method perform in settings where the original model produces much longer reasoning traces?
3. Can you add discussion of the key intuition for why Theorem 3.10 holds and why discounting leads to an efficient policy?
4. Do the theoretical results extend to stochastic sampling, multi-step agents, or real-valued rewards?
5. Can you clarify the training data used?

---

> ### Author Response · Authors · 2025-11-23
>
> We thank the reviewer for their feedback.
>
> Response to (some) weaknesses:
>
> 1. We are not sure what the reviewer means by the phrase “standard long chain-of-thought setting”. Our experimental setup was chosen because there are strong baselines that we can build off of–such as the official algorithm baselines of bytedance’s VERL library [1]–which is mentioned in our manuscript (see lines 403-410). We interpret the reviewer's comment here to mean the reviewer would like us to run experiments with base models that produce longer reasoning traces. Given our already comprehensive theoretical and empirical results–we expect the trend to generalize to base models that produce longer chains of thought–however as we no longer have access to the compute used to produce the original experiments producing these results will take some time.
> 2. Our primary objective is to establish, both theoretically and empirically, that there exists a regime where reasoning length can be reduced without reducing accuracy. Previous works on length penalties generally report a trade-off: length is reduced at the cost of a small drop in accuracy [2, 3, 4, 5]. We therefore focused on the undiscounted baseline to demonstrate that this trade-off is not fundamental. We remark that discounting in the MDPs considered in this work can be viewed as a type of adaptive length penalty. Consider a trajectory $\tau$ that correctly answers a question at timestep $t$. The cumulative discounted reward is $\gamma^t$. To map this to an adaptive length penalty $\rho(\tau)$, we equate the discounted return to a penalized objective formulation $r(\tau)=1$: $\gamma^t = r(\tau)(1 - \rho(\tau))$. Solving for the penalty term yields: $\rho(\tau) = 1 - \gamma^t$. If $\gamma$ is the Blackwell discount factor, then our discounting can be viewed as the theoretically optimal way of applying an adaptive length penalty.
> 3. Thank you for this suggestion. We agree that more intuition would improve accessibility. Theorem 3.10 leverages the following key observation: in this setting, the discounted return for small $\epsilon = 1 − \gamma$ admits the expansion:
> $J_\gamma(\pi) = p(\pi)(1 - \epsilon(L(\pi) - 1)) + O(\epsilon^2),$
> where $p(\pi)$ is the success probability and $L(\pi)$ is the expected length of successful trajectories. As $\gamma \to 1$ ($\epsilon \to 0$), maximizing $J_\gamma$ first prioritizes maximizing $p(\pi)$; among policies with the same $p(\pi)$, it then prefers smaller $L(\pi)$. Blackwell optimal policies, being optimal for all $\gamma$ sufficiently close to 1, must therefore do both. We will include this discussion in the camera ready version of our manuscript
> 4. Our results can be (trivially) extended to stochastic shortest-path (SSP) problems—which have noise in environment transitions–as deterministic transitions were never used in proving the theorem. We will update Theorem 3.10 to reflect this, as its proof only requires deterministic binary terminal rewards, not deterministic transitions. We chose to focus on deterministic transitions initially as this models standard language tasks where token concatenation is deterministic. This SSP framework applies to multi-step agents and tool-calling settings if they can be modeled as finite-horizon MDPs where success is a binary terminal reward. However, our results do not generally hold for arbitrary real-valued reward structures, such as ones where the reward is +1 for every token produced and the notation of a shortest path is to simply produce more tokens. We will include the result on SSP in the camera-ready version for completeness.
> 5. Theorems 3.5 and 3.10 only require a finite restricted policy class $\Pi$, which includes the stochastic softmax policies used during training. Our analysis explicitly covers the common paradigm of training with a stochastic policy and deploying (evaluating) the greedified policy ($\Sigma$). Our experiments employ greedy decoding only during evaluation–and the usual softmax policies for training–since the policy gradient theorem breaks down when learning with deterministic policies--we will make this explicit in the manuscript. This reflects the practical deployment scenario where users expect a single high-quality answer, making Pass@1 the relevant metric, rather than checking multiple outputs.
> 6. We discussed our empirical setup in Section 5 (lines 382-396). We will clarify the details further (see Q5 below) and release our code in the camera-ready version of our work.
> 7. We clarify that in MDPs with binary terminal rewards, discounting can be written as an adaptive length penalty. Our contribution is showing that plain old discounting is a theoretically optimal way to maximize accuracy and minimize average response length and then to demonstrate this finding empirically.

---

> ### Author Response · Authors · 2025-11-23
>
> 1. The main contribution of this work is rigorously establishing that a trade-off between response length and accuracy need not exist, contrary to previous works [2,3,4,5]. The purpose of our experiment is to validate our finding–that accuracy and response length need not be traded-off up to some instance dependent threshold–and propose a simple, scalable method to make use of our theoretical findings in practice. We will attempt to include some comparisons with the methods of [2,4]--once our longer chain of thought experiments finish, though the methods within these works both do not have the right theoretically guarantees and sacrifice accuracy for response length.
> 2. The method performs as the theory predicts on settings where the original model produces longer reasoning traces (namely we are optimizing for a shortest path policy). We have many different models producing a variety of base response lengths (from ~125 on gsm8k to ~1263 on AIME) and for each model on each benchmark, accuracy is preserved while response length decreases. We can always run additional experiments with base models on problems that produce longer responses (given enough time)–but we believe our current set of results establish the trend predicting by our theory on a multitude of model/problem combinations. We are currently running more experiments on models that produce longer reasoning traces.
> 3. Yes. The core intuition is the expansion $J_\gamma(\pi) \approx p(\pi) - (1-\gamma)p(\pi)(L(\pi)-1)$. For $\gamma$ close to 1, maximizing $J_\gamma$ first maximizes success probability $p(\pi)$, and then minimizes expected length $L(\pi)$. We will include this discussion in the camera-ready version.
> 4. Yes, Our results hold for any restricted policy class, including stochastic softmax policies (Theorems 3.4 and 3.10). We will make this clearer by moving the greedy deployment class result to the appendix. Multi-step agents: Yes, provided they fit the stochastic shortest-path structure (finite horizon, binary terminal rewards), as discussed above. Real-valued rewards: In general, no--for example +1 for every action produced.
> 5. As discussed in our manuscript (see lines 392-402), we train on 3 different setups:
>
>         i) Qwen3-1.7B, Qwen2.5-7B, and Llama-3-8B trained on the GSM8K training set and evaluated on the GSM8K test set.
>
>         ii) Qwen2.5-7B and Llama-3-8B trained on the MATH training set and evaluated on the MATH500 dataset.
>
>         iii) Qwen2.5-14B and Phi-4 trained on the DeepScalaR dataset and evaluated on AMC 2023, AIME 2025, MINERVA and OLYMPIAD.
>
> We will link the exact Hugging Face datasets in the camera-ready version--the links are here: https://huggingface.co/math-ai / https://huggingface.co/datasets/openai/gsm8k / https://huggingface.co/datasets/nlile/hendrycks-MATH-benchmark / https://huggingface.co/datasets/agentica-org/DeepScaleR-Preview-Dataset.
>
>
>
> [1] https://verl.readthedocs.io/en/latest/algo/baseline.html
>
> [2] Arora, Daman, and Andrea Zanette. "Training language models to reason efficiently." arXiv preprint arXiv:2502.04463 (2025).
>
> [3] Xiang, Violet, et al. "Just Enough Thinking: Efficient Reasoning with Adaptive Length Penalties Reinforcement Learning." arXiv preprint arXiv:2506.05256 (2025).
>
> [4] Su, Jinyan, and Claire Cardie. "Thinking Fast and Right: Balancing Accuracy and Reasoning Length with Adaptive Rewards." arXiv preprint arXiv:2505.18298 (2025).
>
> [5] Team, Kimi, et al. "Kimi k1. 5: Scaling reinforcement learning with llms." arXiv preprint arXiv:2501.12599 (2025).

---

### Official Review · Reviewer_mEWe · 2025-11-02

**Soundness:** 3
**Presentation:** 3
**Contribution:** 3
**Rating:** 6
**Confidence:** 4

**Summary:**

The  problem addressed in the paper  is that RL post training improves
accuracy but lengthens responses, raising computational cost and latency. It
says that the response can be reduced to a threshold without reducing the
accuracy.

The authors have modeled a verifier-based reasoning as a finite horizon MDP
with determinist transitions and binary terminal reward. And then they train
with discount factor (γ < 1). Here only the environment reward is discounted,
leaving   the   formatting/shaping   reward   undiscounted.   Also,   only   the
reasoning tokens are discounted, regularized with a KL penalty to a moving
reference policy. This ensured that the token budgets comparable.

They show that in a fixed policy class Blackwell optimality policies maximizes
undiscounted   success.   And   among   accuracy   maximizers,   minimize   the
expected trajectory length. They reduce average response length on GSM8K,
MATH and other benchmarks while matching pass@1 base line Using GRPO
with discounted objective with (Group Relative Policy Optimization) with
discounted objective.

When tested on Quen 2.5 7B-Instruct and Lama 3 8B-Instruct models, the
response length was reduced in discounted method. When compared to
undiscounted method the GSM8K dataset showed 22% reduction in response
length while MATH dataset showed 13% reduction. The Phi4 and Qwen 2.5
14B-Instruct models with AMC 2023, AIME 2025, MINERVA and OLYMPIAD
datasets also showed reduction in response length.

**Strengths:**

1)	The four design components used here are Discounting only the environment reward, Regularizing KL to a changing reference, Discounting only reasoning tokens and Comparable token budgets across methods. These components are simple to adopt and are also defended in the theory.
2)	In the finite restricted policy classes, for γ close to 1 the Blackwell optimal policies are accuracy maximizing and have shortest mean response length. This is proved in the theorems 3.4, 3.7 and 3.10 which strengthens the claim.
3)	The GSM8K dataset showing a 22% reduction and MATH datasets showing a 13% reduction in response lengths is significant when considering computational cost and latency.

**Weaknesses:**

1)	Modeling as a finite horizon with deterministic transitions and binary terminal reward worked theoretically. But many real-world reasoning workflows violate these assumptions.
2)	In Blackwell optimality analysis, the γ is selected as for from 1 as possible. However, this is done by a simple bisection search.
3)	Discounting is only applied to reasoning tokens. The authors say that discounting entire response slightly hurt the accuracy. This suggest there may be some issues that prevents errors frond being detected.

**Questions:**

1)	In addition to binary verifier rewards, have you tried Dense reward models or Hybrid rewards?
2)	Instead of a bisection search for the selecting γ, will a predictive hybrid approach provide a better result?
3)	Besides reduction in response length, did the latency and computational cost get reduced?
4)	For GSM8K and MATH datasets you limited the completion length to 786 tokens & 2048 tokens. What happens when the runs hit the limit? Have you tried with larger limits?

---

> ### Author Response · Authors · 2025-11-23
>
> We thank the reviewer for their valuable feedback.
>
> Response to (some) of the reviewers weaknesses:
>
> 1. We adopt this abstraction because it accurately models a large class of current LLM reasoning workflows–such as mathematics or theorem proving which are fundamental and worthy of study on their own. However, the deterministic-transition assumption can be relaxed. As noted in our response to Reviewer id2e, we have (trivially) extended Theorem 3.10 to stochastic shortest-path (SSP) problems with binary terminal rewards, meaning transitions can be stochastic (covering workflows like noisy tool calls).
> 2. The theory requires $\gamma$ to be in the Blackwell region $(\gamma_{bw}, 1)$. Since $\gamma_{bw}$ is unknown in practice, we use bisection search to empirically find the smallest $\gamma$ (farthest from 1) that maintains the undiscounted training accuracy. Much like choosing a learning rate, this search is practical, inexpensive, and numerically stable. It does not require computing the theoretical bound (Theorem 3.9), which is conservative due to its asymptotic nature. We do not see bisection as a weakness–as it chooses a good discount factor exponentially fast– and thus it is a principled and efficient approach.
> 3. The accuracy drop when discounting the entire response is not due to issues with error detection. Rather, it stems from penalizing tokens required to verify success. Verifiers often expect specific formatting (e.g., tags, answer markers). If these formatting tokens are discounted, the model is sometimes incentivized to omit them, leading to outputs that may contain the correct answer but fail the verifier's interface. By discounting only reasoning tokens, we encourage the model to reach the “final reasoning state” (the point where a correct answer can be produced) efficiently, while leaving it free to emit the necessary, undiscounted formatting tokens.

---

> ### Author Response · Authors · 2025-11-23
>
> Response to reviewers questions:
>
> 1. We did not try other reward models, as binary terminal rewards are standard in math reasoning tasks (since an answer can always fall into a satisfactory or unsatisfactory category), and our theory is tailored to this setting. In general, our results will not hold for arbitrary dense rewards. For example, if the reward is +1 for every transition, the optimal behavior is simply to keep producing tokens until termination. Thus, the notion of a shortest-path policy with dense rewards becomes hard to define in general. We are unsure what the reviewer means by hybrid rewards. Some schemes combining binary terminal rewards with dense intermediate rewards might preserve our guarantees, but it depends on the specific instance. If the reviewer could clarify their definition, we would be happy to discuss it further.
> 2. We are unsure what is meant by a “predictive hybrid” approach here, if the reviewer could clarify this approach we would be happy to discuss it further
> 3. Yes–both decreased as the response length decreased.
> 4. We chose these limits (786 for GSM8K, 2048 for MATH, 4096 for DeepScalaR) as we found that our models rarely produced more tokens than these numbers (i.e., responses are rarely clipped for the given prompts). We tested with larger limits but found this did not further improve accuracy--but did incur larger compute costs. As mentioned in the paper (see lines 403-410), our hyperparameters were chosen to match or exceed strong baselines (such as the official VERL algorithmic baselines [1] or those of strong related papers [2]). Instead, we spent our computational budget on more generations and more runs across training seeds.
>
> [1] https://verl.readthedocs.io/en/latest/algo/baseline.html
>
> [2] Roux, Nicolas Le, et al. "Tapered off-policy reinforce: Stable and efficient reinforcement learning for llms." arXiv preprint arXiv:2503.14286 (2025).

---

### Official Review · Reviewer_id2e · 2025-11-03

**Soundness:** 3
**Presentation:** 3
**Contribution:** 3
**Rating:** 6
**Confidence:** 3

**Summary:**

LLM reasoning models generate verbose intermediate reasoning to solve complex problems, incurring high computational costs. While reinforcement learning improves accuracy, it often lengthens responses. This paper challenges the assumption that this accuracy-length trade-off is unavoidable (as discussed in prior works).

The authors apply discounting (γ < 1) to the reward function, motivated by Blackwell optimality from decision theory. They prove that within any fixed policy class, such a policy exists, establishing that no inherent trade-off between accuracy and reasoning length occurs up to an instance-dependent regime. The key practical insight seems to be to discount only the environment (correctness) reward while leaving intrinsic (formatting) rewards undiscounted. Combined with KL regularization and discounting only reasoning tokens, this approach substantially reduces response length without sacrificing accuracy.

To back up the theoretical work, the authors conduct experiments on math benchmarks (GSM8K, MATH, AMC, AIME, MINERVA, OLYMPIAD) and demonstrate 22% length reduction for Qwen2.5 7B and 13% for Llama 3 8B while maintaining pass@1 accuracy, even when controlling for total tokens processed during training.

**Strengths:**

* The paper makes a principled design choice to discount only extrinsic (correctness) rewards while leaving intrinsic (formatting) rewards undiscounted, which is well-justified both theoretically and practically.

* The theoretical exposition clearly communicates Blackwell optimality concepts and the main results, making the mathematical framework accessible (I am not a hard RL person and found it pleasant to read). The paper provides thorough theoretical analysis that comprehensively covers finite policy classes, softmax training, and greedy deployment scenarios.

* The experimental methodology is good (e.g., model selection), with results averaged over multiple seeds to properly control for variance inherent in RL-style post-training. The efficiency claims are validated even when matching token budgets across methods, ruling out the confound that discounted methods simply see less data. Ultimatively, the results consistently support the theoretical predictions across multiple benchmarks and model architectures.

**Weaknesses:**

The paper
* evaluation is limited to math benchmarks (GSM8K, MATH, AMC, AIME, MINERVA, OLYMPIAD); it remains unclear whether these findings generalize to coding tasks or other reasoning domains.

* theory section relies on the deterministic transitions assumption, but the paper does not specify whether training uses stochastic sampling (T>0) or greedy decoding (T=0), creating a potential theory-practice gap.

* assumes the policy class is finite, which is needed for the existence of the Blackwell discount factor, but it is unclear how realistic this assumption is or whether violations would significantly impact the results.

* has no systematic analysis showing where the difficulty cutoff lies, i.e., at what problem hardness the efficiency gains from discounting diminish or disappear.

* applies a single discount factor γ per dataset but does not justify whether this is appropriate or whether different problems within a benchmark should use problem-specific discount factors.

* does not discuss why discounting intrinsic rewards would be problematic, missing an opportunity to strengthen the design justification.

**Questions:**

* How dependent is the "no trade-off" result on the quality of the base model and the difficulty of the problem? Could this approach only be effective for sufficiently large models on sufficiently simple problems and fail to scale as a general training paradigm?

* Why has no one already tested this approach empirically given that discounting seems elegant and theoretically motivated? I am mainly asking as you mention ByteDance's existing implementation (Sheng et al. 2025) that already seems to do something similar, and if so, what exactly is novel here?

* Beyond maintaining pass@1 accuracy, what is the impact on reasoning quality (coherence, correctness of intermediate steps)?

* When problem difficulty increases and discounted RL is applied, does training become less effective or does the mean chain-of-thought length simply increase again, negating efficiency gains?

* In practical deployment across multiple tasks, should one use a universal discount factor or does γ need to be tuned per dataset or per problem?

---

> ### Author Response · Authors · 2025-11-23
>
> We thank the reviewer for their valuable feedback.
>
>
> Response to (some) weaknesses:
>
> 1. Our theoretical analysis is domain-agnostic, requiring only a finite-horizon, verifier-based MDP structure--with stochastic transitions and binary terminal rewards. This framework applies to many domains, including coding tasks where programmatic verifiers (e.g., unit tests) are available. Experimentally, we focused on mathematics primarily because strong verifier-based RL setups and robust baselines (see lines 403-410 of our manuscript) exist for GSM8K and MATH allowing for more rigorous evaluation. We trained on GSM8K, MATH, and DeepScalaR, and evaluated both in-distribution (GSM8K/MATH) and on distinct, more challenging benchmarks (Train on DeepScalaR evaluate on AMC, AIME, MINERVA, OLYMPIAD). Across five different base models (Qwen2.5-7B, Qwen2.5-14B, Qwen3-1.7B, Phi-4, Llama-3-8B), we consistently observe that discounting reduces average reasoning length while maintaining or slightly improving Pass@1.
> 2. The deterministic transition assumption refers to the environment dynamics (not policy/LLM): in language model reasoning MDPs, given a prefix (state) and a token (action), the next state (concatenation) is deterministic–the action given the state can be random. That there is a theory-practice gap is untrue as we train with softmax policy ($\tau=1$) and evaluate the greedy policy ($\tau=0$) which is policy class $\Sigma$ studied in section 3.2--this is detailed in lines 403-405 of our manuscript where we mentioned that we use $\tau=0$ greedy decoding at inference time in line with our Theorem 3.9. We can clarify this to say at evaluation (or test time). Note that $\tau>0$ for training is implicit otherwise we would not be able to use policy gradient algorithms.
> 3. The finite-policy-class assumption is used to guarantee the existence of a Blackwell discount factor $\gamma_{bw}$ and to obtain an explicit bound. This assumption is satisfied for any policy classes that can be represented with finite precision (so the class all bfloat16 finite dimensional vectors). Crucially, the main structural result (Theorem 3.10)—that Blackwell-optimal policies are accuracy-maximizing and shortest-path—holds for any finite restricted policy class $\Pi$ and more importantly our results (all theorems) do not depend on the size of the class. Our results do not require restricting to greedy policies–just arbitrary finite classes which include softmax policies classes for training and deployment. In the revision, we will restructure the paper to emphasize this generality. If one considers finitely many benchmarks simultaneously, they can be treated as a single MDP with a mixed initial-state distribution $\mu$. The same finite $\Pi$ and corresponding $\gamma_{bw}$ apply to this joint problem, making the assumption realistic for practical scenarios.
> 4. We are not sure what the reviewers are trying to say here and, our theory just states that discounting reduces average response length across initial prompts--we do not use any problem difficulty heuristics as reducing response length is done in a data drive manor. If the reviewer could clarify this point we would be happy to engage the reviewer with further discussions.
> 5. We justify the appropriateness of using a single $\gamma$ by establishing this is sufficient for finding shortest path policies with response to a distribution over prompts (problems within a desired set of benchmarks). So long as the distribution over prompts covers most of the desired benchmarks then it is appropriate to use a single $\gamma$ as state in our Theorem 3.10.

---

> ### Author Response · Authors · 2025-11-23
>
> 1. As mentioned in our manuscript, the trade-off result depends on both the model class (the class of language models being optimized) and the class of MDPs (the distribution of prompts), as these determine the Blackwell discount factor. We have no reason to believe this approach will not scale as a general training paradigm. Across five different base models (Qwen2.5-7B, Qwen2.5-14B, Phi-4, Llama-3-8B, and Qwen3-1.7B), three training sets (GSM8K, MATH, DeepScalaR), and six evaluation sets (GSM8K, MATH, AMC, AIME, MINERVA, OLYMPIAD), we consistently see that we can reduce average response length while maintaining accuracy (both in-distribution train GSM8K/MATH eval on GSM8K/MATH and out of distribution train DeepScalaR and eval on AMC/AIME/MINERVA/OLYMPIAD). Our theory is intended to cover the cases for which we did not have the compute to evaluate.
> 2. While standard RL libraries (e.g., ByteDance VERL) support discounting, our novel contribution is the connection to Blackwell optimality in the context of LLM reasoning. This connection reveals crucial insights: a) The required discount factors are much closer to 1 (e.g., $1 − 10^{-7}$) than typically used in RL (e.g., 0.9/0.99) [1], as suggested by our bound (Theorem 3.9). Anecdotally, the RL practitioners we spoke to rarely think to set the discount factor this high–and were pleasantly surprised by our Blackwell explanation. b) Discounting must be implemented/evaluated with care: applied only to extrinsic (correctness) rewards and over reasoning tokens, leaving intrinsic (formatting) rewards undiscounted and also using comparable training budgets. Prior work using length penalties often reported an accuracy drop when shortening responses [2,3,4,5]. Our work is the first to rigorously establish, theoretically and empirically, that there is an instance-dependent regime where this trade-off need not occur.
> 3. We qualitatively examined samples across benchmarks and did not observe a degradation in logical coherence or the correctness of intermediate steps. Often, the discounted model produced a shorter explanation by removing redundant reasoning rather than essential steps. We agree that a systematic analysis is valuable. In the camera-ready version, we will include (i) selected qualitative side-by-side examples, and (ii) results from an external LLM-as-a-judge evaluation (e.g., using Gemini AutoRater) scoring the coherence and correctness of intermediate steps.
> 4. We ran experiments across varying difficulties. When problems become harder (e.g., moving from GSM8K to AIME), both undiscounted and discounted policies generate longer chains of thought. However, the efficiency gains are not negated. The discounted policies remain consistently shorter than their undiscounted counterparts while matching Pass@1. For example, we see substantial length reductions on the harder AMC/AIME/MINERVA/OLYMPIAD benchmarks (Table 2). Thus, discounted training remains effective; the accuracy-length frontier shifts, but our method continues to select relatively shorter successful trajectories.
> 5. Our analysis suggests that a universal $\gamma$ should be chosen based on the policy class $\Pi$ and prompt distribution $\mu$. In practice, this means choosing $\gamma$ for a given model and a desired distribution over tasks (maybe uniform distribution over math/coding/science questions for example).
>
> [1] Sutton, Richard S., and Andrew G. Barto. "Reinforcement learning: an introduction, 2nd edition" (2018).
>
> [2] Arora, Daman, and Andrea Zanette. "Training language models to reason efficiently." arXiv preprint arXiv:2502.04463 (2025).
>
> [3] Xiang, Violet, et al. "Just Enough Thinking: Efficient Reasoning with Adaptive Length Penalties Reinforcement Learning." arXiv preprint arXiv:2506.05256 (2025).
>
> [4] Su, Jinyan, and Claire Cardie. "Thinking Fast and Right: Balancing Accuracy and Reasoning Length with Adaptive Rewards." arXiv preprint arXiv:2505.18298 (2025).
>
> [5] Team, Kimi, et al. "Kimi k1. 5: Scaling reinforcement learning with llms." arXiv preprint arXiv:2501.12599 (2025).

---

> > ### Comment · Reviewer_id2e · 2025-11-26
> >
> > Thank you for your detailed response and clarifications! With the promised systematic analysis, my concerns have essentially been addressed or have been resolved via clarifications. I've increased my score accordingly (6 -> 8), but will keep my certainty score (3), as the revisions are promised for the camera ready and will not be included in a rebuttal revision of the submission.

---

> > > ### Author Response · Authors · 2025-11-27
> > >
> > > Thank you, we are happy that you found our theory pleasant to read :)

---

### Author Response · Authors · 2025-11-28

Are we to response only to the reviewers original review or should I also take into account comments made after the rebuttal period began? This is in response to the message that was sent from the PCs are reviews/scores being revert to their pre-rebuttal forms.

---

> ### Author Response · Authors · 2025-12-03
>
> We have included a revised version of our paper that addresses some of the common concerns of our work.
>
> 1. We have relegated most of the theory and discussion of the greedy deployment class to our appendix and included a brief discussion about the class at the end of Section 3.
>
> 2. We have removed the assumption that transitions must be deterministic in Theorem 3.10 and show that the result holds under only a binary terminal reward.
>
> 3. We have included a larger discussion on the intuition of our theory. Specifically, we included a new result showing why Blackwell optimality gives rise to a shortest path policy when considering restricted policy classes—such as the softmax policy class—and included a proof sketch of Theorem 3.10.
>
> 4. We have lightly edited the problem setting section (Section 2) to make it more explicit that we consider general finite-horizon MDPs and included a subsection describing language model reasoning in the context of MDPs.
>
> We thank the reviewers for their helpful feedback in enhancing the presentation of our results.

---

### Meta-Review · Area_Chair_ciL4 · 2026-01-05

**Summary:**

This paper studies reasoning in large language models, with the primary goal being to offer a method that minimizes token usage while maintaining response accuracy. The paper leverages the Blackwell optimality theory and establishes that, under some assumptions, no inherent trade-off between accuracy and reasoning length occurs up to an instance-dependent regime. The theoretical finding is empirically supported on math benchmarks (GSM8K, MATH, AMC, AIME, MINERVA, OLYMPIAD).

The reviewers appreciated the practical relevance and the motivation behind the problem formulation, as well as consistent empirical superiority across models. They raised some concerns, summarized next, most of which are sufficiently addressed by the rebuttal. Overall, given the status of the rebuttal and revised manuscript, I support the paper by recommending acceptance.

**Reviewer Concerns:**

The most important concerns raised by the reviewers include:
- __The evaluation is limited to math-type benchmarks.__ As adequately discussed in the rebuttal, the method is domain-agnostic, requiring a a finite-horizon and a terminal reward. Even though it limits the scope, it still encompasses a broad range of tasks.
- __Reliance on deterministic transitions.__ This is fully addressed in the rebuttal. The assumption is lifted and the result holds under only a binary terminal reward.
- __The assumption of finite policy class.__ As the rebuttal explains, it is indeed needed to guarantee the existence of a Blackwell discount factor and to obtain an explicit bound. It can be relaxed in practical situations.
- __Sparse or Dense Rewards?__ As discussed in the rebuttal, the method does not support dense rewards. More precisely, the proposed theory only supports binary terminal rewards.
- __Using bisection to determine the Blackwell discount factor and its theoretical implications.__ It was adequately discussed in the rebuttal.
- __Support of stochastic sampling, multi-step agents, or real-valued reward__ Some settings of multi-step agents are covered by the method. However, it does not generally support arbitrary real-valued reward structures.
- __Clarifications and questions about empirical evaluations.__ Addressed in the rebuttal.
- __Concerns about technical novelty and usefulness of the presented bounds.__ The rebuttal clarifies these points.

**Reviewer Scores:**

- Reviewer id2e raised some concerns. I believe they are well addressed in the rebuttal, and the reviewer already expressed willingness to increase it to 8.
- Reviewer mEWe: The reviewer is readily positive. The rebuttal addressed the raised questions, so it is not unlikely that the score is slightly increased.
- Reviewer GRgL: The comments are well addressed by the rebuttal, so I would see it likely that the reviewer increased the score to 6.
- Reviewer Dg1G: The reviewer raised some key concerns that were addressed by the rebuttal. Although the responses render proper and offer adequate explanations and clarifications, the reviewer seemed unwilling to increase the score, claiming the paper needs a major revision to implement the changes. However, if I were to step in as a reviewer, I would increase the score.

---

### Decision · Program_Chairs · 2026-01-26

Accept (Poster)